# OUT-OF-DISTRIBUTION GRAPH MODELS MERGING

**Yidi Wang**[1,2]   **Ziyue Qiao**[1*]   **Jiawei Gu**[1]   **Xubin Zheng**[1]   **Pengyang Wang**[2]
**Xiaobing Pei**[3]   **Xiao Luo**[4]
[1]School of Computing and Information Technology, Great Bay University
[2]The State Key Laboratory of Internet of Things for Smart City, Department of Computer and
Information Science, University of Macau
[3]School of Software Engineering, Huazhong University of Science and Technology
[4]Department of Statistics, University of Wisconsin–Madison
`wang.yidi@connect.um.edu.mo`   `ziyuejoe@gmail.com`

## ABSTRACT

This paper studies a novel problem of out-of-distribution graph models merging, which aims to construct a generalized model from multiple graph models pre-trained on different domains with distribution discrepancy. This problem is challenging because of the difficulty in learning domain-invariant knowledge encoded implicitly within model parameters and consolidating expertise from potentially heterogeneous GNN backbones. In this work, we propose a graph generation strategy that instantiates the mixture distribution of multiple domains. Then, we merge and fine-tune the pre-trained graph models via a MoE module and a masking mechanism for generalized adaptation. Our framework is architecture-agnostic and can operate without any source/target domain data. Both theoretical analysis and experimental results demonstrate the effectiveness of our approach in addressing the model generalization problem. The code is available at `https://github.com/siriuslay/OGMM`.

## 1 INTRODUCTION

As the scale and complexity of observed graph data continue to increase, graph models have become essential tools for extracting insights from real-world scenarios (Zhu et al., 2022; Li et al., 2022; Zhang et al., 2024; Wu et al., 2024a). Recently, Graph Model Generalization (GMG) aims to transcend the limitations of multi-domain datasets with distribution shifts by identifying invariant features (Arjovsky et al., 2019; Chang et al., 2020; Ahuja et al., 2021), causal relationships (Gui et al., 2024; Chen et al., 2024a), or risk extrapolation (Xu et al., 2020; Ziyin et al., 2020; Ye et al., 2021; Li et al., 2024c) underlying the graph data distributions. The objective is to maintain robust performance on unseen, out-of-distribution graphs.

Current research focuses on training a generalized model from scratch using graph data from multiple domains with distribution discrepancy. However, a less explored yet practical scenario emerges when graph models have already been trained individually on these different domains–referred to as *Out-of-Distribution Graph Models*. For instance, in social networks, models trained on user data from different groups or with varying architectures capture diverse behavior patterns. Achieving a unified and generalized model on these datasets usually needs training from scratch, which is complex and wasteful of their learned knowledge.

As presented in Figure 1, these models are designed for similar tasks but on different datasets, each preserving specialized knowledge. Figure 2 illustrates the performance of GNN models pre-trained on one domain and tested on both their own and other domains with distribution shifts (detailed setting is in Sec. 4.1). While models perform well in their own domain, their performance degrades in others, and different GNN architectures may excel in different domains. These suggest that by merging these models' intrinsic invariance and complementary expertise, it is possible to address challenges arising from distribution shifts and achieve generalization on all domains, even without retraining from scratch on the original training datasets or labels.

---

*Corresponding Author.

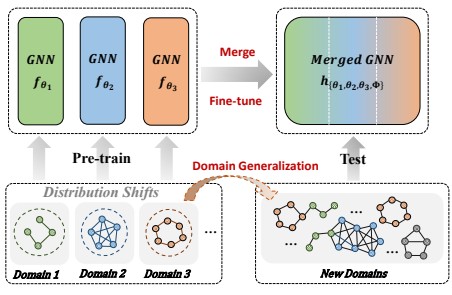

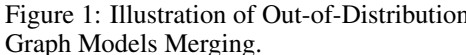

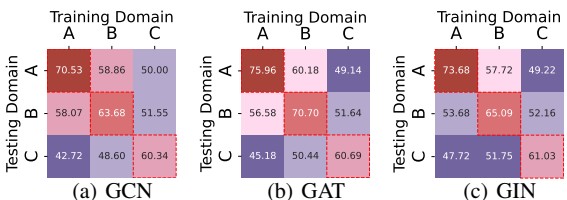

Figure 1: Illustration of Out-of-Distribution Graph Models Merging.

Figure 2: Comparison of different GNN models' generalization performance on PTC between in-distribution and OOD scenarios, with three domains represented as A / B / C. Values indicate Acc (%). The results within the red dashed box represent the best performance.

Therefore, this paper investigates a novel and practical problem, named *Out-of-Distribution Graph Models Merging*: How to consolidate the knowledge of multiple pre-trained GNNs into a unified model that generalizes under distribution shifts? Achieving this goal is non-trivial due to the following challenges: (1) Unlike conventional domain generalization approaches, learning the domain-invariant knowledge from the domain data explicitly, learning from the model parameters in our setting is inherently complicated. (2) Furthermore, the pre-trained models may differ in their architectures and hyperparameters, making it difficult to consolidate the expertise of these diverse models into a unified representation.

To address these challenges, we propose a novel Out-of-distribution Graph Models Merging (OGMM) framework for domain generalization, which is depicted in Figure 3. Specifically, we explore the theory of multi-domain generalization defining generalization risk in functional space and deriving a two-stage objective function. The first stage is a domain knowledge generation process. We "invert" each pre-trained GNN (expert), to generate a small set of label-conditional graphs starting from random noise. These generative graphs are then aggregated as the training data for the second stage. The second stage involves experts fine-tuning and merging. To effectively retain the source domain knowledge learned by models with different parameters and architectures, we employ a Mixture-of-Experts (MoE) module for merging. Meanwhile, based on the mixture distribution assumption, we prove that the fine-tuned MoE with masks serves as an approximation of the generalization risk function. The lightweight sparse gating weights and the masked experts are trained with the generative graphs, enabling the allocation logic of "sample-expert" pattern. The main contributions of OGMM are summarized as follows:

- We propose a novel framework named out-of-distribution graph models merging, which aims to learn a generalized model from multiple graph models pre-trained under domain shifts.

- We propose a graph generator for concentrating the model knowledge effectively, and develop an innovative model merging function utilizing fine-tuned MoE to address adaptive integration of multiple pre-trained models, thereby enhancing generalizability to unseen graphs.

- We validate OGMM on various tasks, demonstrating substantial improvements on out-of-distribution data compared to both individual model and traditional model merging methods.

## 2 PROBLEM FORMULATION

**Graph Neural Networks (GNNs).** A graph is represented as $G = \{A, X\}$, where $A \in \mathbb{R}^{n \times n}$ is the adjacency matrix and $X \in \mathbb{R}^{n \times d}$ denotes the node features, with $n$ being the number of nodes in $G$. We consider a basic GNN consisting of two parts: $\{\Psi, \Phi\}$, *i.e.*, $f(\Theta) = \theta_\Psi \circ \theta_\Phi \to \mathcal{Y}$, where $\theta_\Psi$ is parameters in the graph encoder, $\theta_\Phi$ corresponds to the classifier parameters, and $\mathcal{Y}$ is the graph-level (or node-level) label space in the downstream tasks. Specifically, $\Psi$ represents a multi-layer message aggregation function, where the update mechanism in the $L$-th layer can be written as follows:

$$h_i^{L+1} = \sigma(\text{AGGR}(h_i^L, \{h_v^L | v \in \mathcal{N}(i)\})), \tag{1}$$

where $h_i^0 = x_i$, and $h_i^L$ is the output representation for node $i$. $\sigma$ is an activation function. $\text{AGGR}(\cdot)$ defines the aggregation of nodes and their neighbors $\mathcal{N}$. The classifier $\Phi$ will be trained to assign a label for each graph (or node) from the label space $\mathcal{Y} = \{Y_1, Y_2, \dots, Y_c\}$ with $c$ classes.

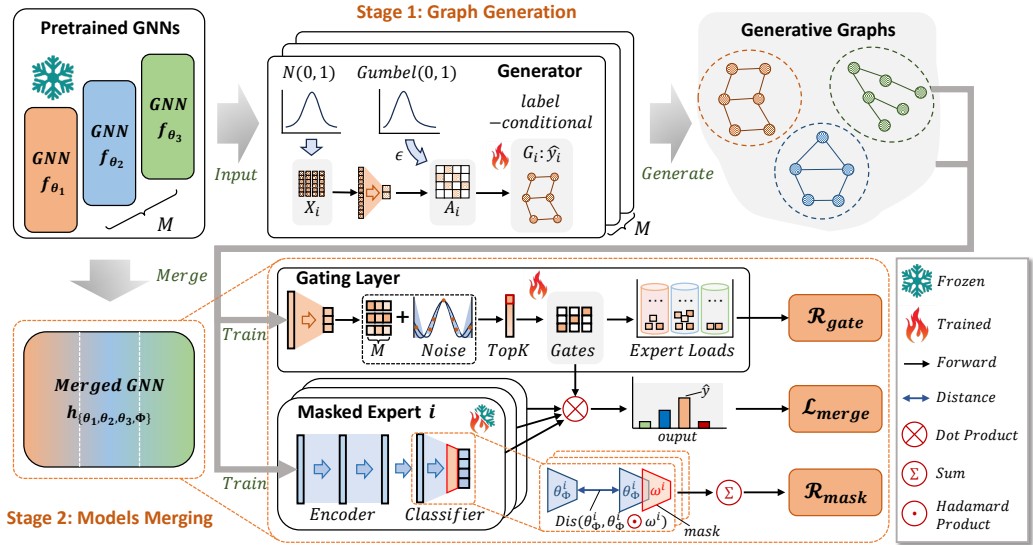

Figure 3: Architecture overview. The architecture of OGMM consists of two primary stages: (1) Graph generation. Each pre-trained GNN serves as a supervisor to train its corresponding generator, which reconstructs label-conditional graphs from random noise. (2) Model merging. The generative graphs are aggregated to train a merged GNN using a MoE module. It comprises a gating layer and a set of fine-tuned masked experts. Gradient updates are guided by mask and gating regularization terms alongside classification loss.

**Out-of-distribution Generalization on Graphs.** The objective of Out-of-distribution Generalization (also known as multi-domain generalization) is to leverage joint data samples from multiple source domains to capture cross-domain invariant knowledge (Crammer et al., 2008; Mansour et al., 2008). Here, we present its formulation in the context of graph domains. Suppose we are given $M$ sets of source data, denoted as $\{\mathcal{G}_i\}_{i \in M}$, where $\mathcal{G}_i = \{G_1, G_2, \ldots, G_{N_i}\}$ represents the $i$-th source dataset. Each $\mathcal{G}_i$ maps to the label space $\mathcal{Y}$. Additionally, we are provided with a target dataset consisting of $N_t$ graphs ($N_t = 1$ for node-level tasks), $\mathcal{G}_T = \{G_1, G_2, \ldots, G_{N_t}\}$, which shares the same label space $\mathcal{Y}$ as the source data but follows different distributions. The goal is to optimize a GNN model $f(\cdot)$ with parameter $\Theta$ from scratch to minimize the generalization error under the unseen shifts as:

$$f(\Theta^*) = \arg \min_{\Theta} \mathbb{E}_{\mathcal{G}_T, \mathcal{Y}} \big[ \ell(f(\Theta, \{\mathcal{G}_1, \mathcal{G}_2, ..., \mathcal{G}_M\}), \mathcal{G}_T, \mathcal{Y}) \big]. \tag{2}$$

**Out-of-distribution Graph Models Merging.** Different from the conventional conditions of Out-of-distribution Generalization, Out-of-distribution Models Merging assumes that the task-specific GNNs $\{f(\Theta_i)\}_{i \in M}$ have already been trained on different datasets $\{\mathcal{G}_i\}_{i \in M}$ and aims to learn a unified model utilizing the parameters of multiple pre-trained models. The objective is to optimize a multi-model merging function to obtain a model with higher generalization capabilities. Under the proposed *Graph Models Merging* setting, we define an objective function as follows:

$$\Gamma^* = \arg \min_{\alpha} \mathbb{E}_{\mathcal{G}_T, \mathcal{Y}} \big[ \ell(\Gamma(\alpha, \{\Theta_1, \Theta_2, \ldots, \Theta_M\}), \mathcal{G}_T, \mathcal{Y}) \big], \tag{3}$$

where $\Gamma^*$ is the expected model merging function, $\alpha$ is the combining weights, and $\ell(\cdot)$ is the loss function that measures the prediction error.

## 3 METHODOLOGY

In this section, we present a framework for out-of-distribution graph model merging that constructs a unified model from multiple pre-trained GNNs without access to original data. Building on multi-domain generalization theory, we develop a two-stage approach that addresses the fundamental challenge of extracting and consolidating domain-invariant knowledge from model parameters.

## 3.1 Overall Framework

Here we justify Eq. 3 based on multi-domain out-of-distribution generalization theory, enabling out-of-distribution models merging. To begin, we establish a mixture distribution assumption for this problem, stating that the target distribution is a mixture of distributions from multiple sources.

**Assumption 3.1** (Mixture Distribution). The input to the problem is the set of $M$ source distributions, denoted as $\{\mathcal{G}_1, \mathcal{G}_2, \ldots, \mathcal{G}_M\}$. The distribution of the target domain, $\mathcal{G}_T$ is assumed to be a linear combination of the $M$ source distributions: $\mathcal{G}_T = \sum_i^M \alpha_i \mathcal{G}_i$.

This assumption is widely accepted in multi-domain generalization problems (Crammer et al., 2008; Mansour et al., 2008), and leads to the rule of linear combination of functions, expressed as $\Gamma = \sum_i^M \alpha_i f(\Theta_i)$. Next, we provide the definition of $\mathcal{H}\Delta\mathcal{H}$-divergence to define the symmetric difference in hypothesis space $\mathcal{H}$.

**Definition 3.2.** [$\mathcal{H}\Delta\mathcal{H}$-divergence]. Let $\mathcal{H}$ be a hypothesis class. $f(\Theta_i), f(\Theta_j) \in \mathcal{H}$ are the functions trained on distributions $\mathcal{G}_i$ and $\mathcal{G}_j$, respectively. We define the divergence between $\mathcal{G}_i$ and $\mathcal{G}_j$ in the function space:

$$d_{\mathcal{H}\Delta\mathcal{H}}(\mathcal{G}_i, \mathcal{G}_j) = 2 \sup \left| E_{G \sim \mathcal{G}_i}[|f(\Theta_i, G) - f(\Theta_j, G)|] - E_{G \sim \mathcal{G}_j}[|f(\Theta_i, G) - f(\Theta_j, G)|] \right|. \quad (4)$$

By the linear assumption and the definition of divergence, we prove the generalization error bound of $\Gamma$ on the target distribution $\mathcal{G}_T$ in the following theorem.

**Theorem 3.3.** *If each $f(\Theta_i)$ is an optimal learner trained on the marginal distribution $\mathcal{G}_i$, the upper bound of the generalization error for $\Gamma(\cdot)$ on the target domain is given by the sum of the cross-validation errors of these sub-learners across different distributions.*

The proof is shown in Appendix A.1 due to the page limit. To enhance the generalization capability of the $\Gamma$, we can introduce fine-tuning weights $\omega^i$ for $f(\Theta_i)$ to decrease the cross-validation error. The overall objective for the merging function can be formulated as:

$$\arg \min_{\Gamma} \sum_i^M \left[ \mathcal{C}_{G \sim \mathcal{G}_i}(f(\Theta_i, G), \Gamma(\alpha, \omega, G)) \right] + \sum_i^M \epsilon_i(f(\Theta_i)) + \sum_{i,j}^M \epsilon_j(f(\Theta_i, \omega^i)) + \kappa, \quad (5)$$

where $\epsilon_i(\cdot)$ denotes the empirical error on $\mathcal{G}_i$. $\mathcal{C}(\cdot)$ is a loss function like cross-entropy. $\kappa$ represents the minimum sum of errors achievable by the optimal hypothesis $h$ across all domains within our hypothesis class $\mathcal{H}$. This value is determined by the design of $\mathcal{H}$ itself (like neural network architecture) and is independent of our optimization over $\Gamma$.

Then, we consider the expansion of $\epsilon_i(f(\Theta_i))$ as a starting point for knowledge extraction from $f(\Theta_i)$. Consequently, Optimization 5 can be reformulated as a two-stage objective function:

$$\arg \min_{\alpha, \omega, \mathcal{G}^*} \sum_i^N \mathcal{C}_{G_i \sim \mathcal{G}^*}(\hat{y}_i, \sum_j^M \alpha_j f(\Theta_j, \omega^j, G_i)) \ \ [Sec.\ 3.3]$$

$$\text{s.t. } \mathcal{G}_i^* = \arg \min_{\mathcal{G}_i} \sum_j^{N_i} \mathcal{C}_{G_j \sim \mathcal{G}_i}(\hat{y}_j, f(\Theta_i, G_j)) \ \ [Sec.\ 3.2],$$

$$(6)$$

where $\hat{y}_i$ are the conditional labels sampled from the label space for samples on $G_i$. $\mathcal{G}^* = \sum_i^M \alpha_i \mathcal{G}_i^*$ is the mixture distribution generated from pre-trained GNNs, which will be introduced in Sec. 3.2. The $N_i$ represents the number of samples drawn from $\mathcal{G}_i$. We use these generative samples to fine-tune $\alpha$ and $\omega$ in merging function $\Gamma$, which will be introduced in Sec. 3.3. The analysis details are provided in Appendix A.3. This theorem shows that under the mixture distribution assumption, the generalization ability of the merged GNN depends on three factors: the pre-training error of each model, the fine-tuning error of these models on the new domains, and the training error of the merged model on the generated samples. Next, we will provide the detailed implementations of OGMM.

## 3.2 Label-Conditional Graph Generation

In the first stage, we use pre-trained graph models to generate synthetic graphs for subsequent fine-tuning and merging. Instead of using the original graphs, we opt for generated graphs because:

the original datasets may not always be accessible for every model, generating a smaller set of graphs is more efficient than using the entire dataset, and the generated graphs may sometimes distill and refine knowledge more effectively, making them more representative than the original data. Still, our method is capable of utilizing the original data, and a comparison is provided in Table 2.

As defined by Optimization 6, the goal is to fix all parameters of the pre-trained GNN while optimizing the inputs to minimize the label-conditional posterior error. For graph data, a unique challenge arises due to the inputs' composition of both node features $X$ and graph structure $A$, with $A$ often represented as a discrete variable. This discreteness hinders the direct application of inversion technique (Zagoruyko & Komodakis, 2016; Yin et al., 2020). To address this, Deng & Zhang (2021) employs a discrete gradient approximation method tailored for optimizing $A$. Zhuang et al. (2022) parameterizes $X$ alone while constructing $A$ from the inner product space of $X$, thus preserving feature similarity. Better methods like (Liu et al., 2022; Gao et al., 2024; Jin et al., 2021) use edge encoders to retain inherent relationships between node features and edges. Here, we propose a discrete edge encoder to handle graph structures.

**Graph Generator.** Specifically, for each pre-trained GNN $f(\Theta_i)$, we construct a generator $\mathcal{P}_i$ to produce label-conditional graphs that maximize $f(\Theta_i)$'s agreement. $\mathcal{P}_i$ samples feature $X^i \in \mathbb{R}^{n_i \times d}$ from a standard normal distribution, representing $n_i$ generative nodes, as the initial input for every graph $G_i$. For each $X^i$, $\mathcal{P}_i$ samples a label $\hat{y}_i$ from a uniform distribution, serving as the conditional posterior ground-truth. To generate $A^i$ from $X^i$, we introduce an edge encoder defined as follows:

$$A^i_{jk} = \sigma(\text{MLP}_\theta([X^i_j; X^i_k])), \tag{7}$$

where $\text{MLP}_\theta$ is a three-layer fully-connected neural network, $\sigma$ is an activation function, and $[\cdot; \cdot]$ denotes the concatenation operator. To enforce discrete edge weights, we assume edges follow a Bernoulli distribution and employ the Gumbel-Softmax to approximate values in $[0, 1]$:

$$A^i_{jk} = \text{softmax}\left(\frac{\log(A^i_{jk}) + \mu}{\tau}\right), \tag{8}$$

where $\mu = -\log(-\log(e))$ and $e \sim \text{Uniform}(0, 1)$. Here, $\tau$ denotes the temperature hyperparameter, with $\tau \to 0$ leading $A^i_{jk}$ toward a binary value. By feeding batches of generated samples $(X^i, A^i, \hat{y}_i)$ into the generator $\mathcal{P}_i$, we can use the label-conditional posterior loss $\mathcal{C}(\hat{y}_i, f(\Theta_i, X^i, A^i))$ to fit the source domain distribution, obtaining $\mathcal{G}^*_i$.

**Regularizers for Generation.** In addition to the label-conditional posterior loss, we leverage priors stored in the batch normalization (BN) layers of the pre-trained models. Following (Deng & Zhang, 2021), we enforce the mean and variance values of the generative graph embeddings to match those recorded in the BN layers of the GNNs. Common GNN models perform well with relatively few layers, and correspondingly have a limited number of BN layers (a 2-layer GCN or GAT model typically has only one BN layer while GIN has two). We utilize all BN layers from GNN models to calculate this regularization term:

$$\mathcal{R}_{\text{bn}} = \sum_L \left\{ \left\| \mu_L(\hat{X}^i) - \mathbb{E}\left[ \mu_L(X^i) \mid \mathcal{X}^i \right] \right\|_2 + \left\| \sigma^2_L(\hat{X}^i) - \mathbb{E}\left[ \sigma^2_L(X^i) \mid \mathcal{X}^i \right] \right\|_2 \right\}, \tag{9}$$

where $\hat{X}^i$ denotes the intermediate representations of a graph in the BN layers, while $\mathcal{X}^i$ is the data memorized during training BN layers. $\mu_L, \sigma^2_L$ denote the feature means and variances, respectively, obtained from the $L$-th BN layer.

Another regularization term is the model's confidence in classifying the generative graphs, which ensures that graphs are well-calibrated rather than remaining in an ambiguous state. We define the confidence regularization as follows:

$$\mathcal{R}_{\text{conf}} = \mathbb{E}_{G_i \sim \mathcal{G}^*_j} \left[ -\sum_i^{N_j} f(\Theta_j, G_i) \log f(\Theta_j, G_i) \right], \tag{10}$$

where $\mathcal{G}^*_j$ is the data generated by the $j$-th generator and $N_j$ is the number of samples. Consequently, the overall loss function for each generator is formulated as follows:

$$\mathcal{L}_{\text{gen}} = \sum_{G_i \in \mathcal{G}^*_j} \mathcal{C}(\hat{y}_i, f(\Theta_j, G_i)) + \mathcal{R}_{\text{bn}} + \mathcal{R}_{\text{conf}}. \tag{11}$$

With the parameterized $X$ and $\theta$ learned from the above loss, we can synthesize samples (graphs in graph-level tasks or nodes in node-level tasks) that well-represent the corresponding task data. This process ensures that each generative graph retains structural and feature integrity, without introducing the complexity of gradient approximation methods. Finally, we merge all generated samples to construct the dataset $\mathcal{G}^* = \{\mathcal{G}_1^*, \mathcal{G}_2^*, ..., \mathcal{G}_M^*\}$ for training the model merging function.

### 3.3 Models Fine-tuning and Merging

In the second stage, we need to find a solution for reusing and integrating heterogeneous GNN backbones. This solution should both fine-tune each pre-trained GNN (expert) to adapt to knowledge from multiple domains and be universally applicable to arbitrary model architectures. The objective function for this stage is rewritten according to Optimization 6:

$$\arg\min_{\omega,\alpha} \sum_i^N \mathcal{C}_{G_i \in \mathcal{G}_j^*}(\hat{y}_i, \sum_j^M \alpha_j f(\Theta_j, \omega^j, G_i)), \tag{12}$$

where $N$ and $M$ denote the numbers of samples and models. $\hat{y}_i$ is the generated label of the $i$-th sample, and $\alpha_j$ is the fusion weight, combining different models for different samples, respectively. Indeed, Optimization 12 corresponds to an innovative fine-tuned MoE architecture with Gate Layer ($\alpha$) and added masks ($\omega$). The module's capability is to fine-tune, filter and combine pre-trained experts on a mixture distribution to reach a wider generalization plane overall.

**Masked Experts.** Inspired by mask tuning techniques (Ghanbarzadeh et al., 2023; Li et al., 2024a), we aim to identify and re-weight the pre-trained parameters required by new tasks. Given parameters $\theta_*^i = (\theta_1^i, \ldots, \theta_l^i)^T \in \Theta_i$ of a trained GNN ($l$ is the size of subset in module ($*$)), the mask matrix $\omega^i$ can be optimized as a downstream-related neural pathway:

$$\hat{\theta}_*^i = \theta_*^i \odot \omega^i, \tag{13}$$

where $\odot$ denotes Hadamard product, and $\hat{\theta}_*^i$ replaces $\theta_*^i$ in each Masked Expert. According to Optimization 5, $\Gamma$ represents a distribution-sensitive function, while the role of $\omega^i$ is to fine-tune $f(\Theta_i)$ to minimize $\epsilon_j(f(\Theta_i, \omega^i))$. In shallow networks such as 2-layer GNNs, the position where the mask is added becomes particularly critical. We hypothesize that the weights in the classification head are closely related to downstream tasks, making the model highly susceptible to learning domain-specific knowledge from high-dimensional representations. Thus, fine-tuning the parameters of the classification head is a reasonable and effective strategy, which is further validated by the experimental results provided in Sec. 4.2 and Appendix C.2.

**Sparse Gate in MoE.** Note that we can directly replace $\alpha$ in Optimization 12 with a regular MoE Gate layer, which can be written as follows:

$$\hat{H}_i = \sigma(\sum_{j=1}^M (Gate(X^i)_j H_{i,j})), \tag{14}$$

where $\sigma$ is an activation function, $M$ denotes the number of models (or experts), and $Gate(\cdot)$ is employed to distribute samples to different models. $\hat{H}_i$ and $H_{i,j}$ are the outputs of MoE and the $j$-th pre-trained model, respectively, with respect to sample $X^i$. For all the masked pre-trained GNNs, the sparse gating strategy is as follows:

$$Gate(G_i) = softmax(TopK(Q(G_i), k)), \tag{15}$$

$$Q(G_i) = G_i W_g + \epsilon \cdot softplus(G_i W_n), \tag{16}$$

where $G_i \in \mathcal{G}^*$ is generated from pre-trained GNNs. $TopK(\cdot, k)$ is a selector to find the first $k$ largest (or smallest) members in the sequence. $W_g$ and $W_n$ in Eq. 16 are the learnable weights. $W_g \in \mathbb{R}^{d \times M}$ processes clean sample features to get expert selection scores, while $W_n \in \mathbb{R}^{d \times M}$ adds controlled Gaussian noise $\epsilon \in \mathcal{N}(0, 1)$ to prevent experts from collapsing and ensure load balancing.

Summarizing the above, the loss of Optimization 12 can be re-written as follows:

$$\mathcal{L} = \sum_{G_i \in \mathcal{G}^*} \mathcal{C}(\hat{y}_i, \Gamma_{\omega, W_g, W_n}(G_i)), \tag{17}$$

where $\Gamma_{\omega, W_g, W_n}(G_i) = \sum_{j=1}^{M} Gate(G_i)_j f(\Theta_j, \omega^j, G_i)$ is our proposed model merging function.

**Regularizers for Fine-Tuned MoE.** Here we introduce two regularizers to constrain the optimization direction of gates and masks. Following the strategy in (Wang et al., 2024), we utilize an importance loss to prevent single-selection collapse:

$$\mathcal{R}_{\text{gate}} = CV\left(\sum_{G_i \in \mathcal{G}^*} (Gate(G_i))\right)^2, \tag{18}$$

where $CV(\cdot)$ represents the coefficient of variation. This regularizer measures the degree of weight disparity in "sample-expert" pairings, encouraging uniform weight distribution and enforcing all experts to be "load-balanced". For masks added to the pre-trained GNNs, it is necessary to minimize changes to the frozen parameters while learning new knowledge to prevent "forgetting" old knowledge. So we design the other regularizer as follows:

$$\mathcal{R}_{\text{mask}} = \sum_{i,j} \mathcal{C}_{G_i \in \mathcal{G}^*}(\hat{y}_i, f(\Theta_j, \omega^j, G_i)) + \sum_j \left(\left(\frac{\mathbf{1}^T \cdot \omega^j}{|\omega^j|} - \gamma_v\right) + \left(\frac{\sum_{k:|\omega^{j,k}-1|<\gamma_v} \mathbf{1}}{|\omega^j|} - \gamma_p\right)\right) \tag{19}$$

where $\gamma_v, \gamma_p \in [0, 1]$ are two thresholds to control the effects of the masks in terms of their mean values and variances, respectively. $|\omega^j|$ means the size of $\omega^j$. The first part in Eq. 19 is for learning new knowledge from $\mathcal{G}^*$ and the second part for controlling the process of fine-tuning. The overall loss function for merging is formulated as follows:

$$\mathcal{L}_{merge} = \sum_{G_i \in \mathcal{G}^*} \mathcal{C}(\hat{y}_i, \Gamma_{\Phi}(G_i)) + \lambda_{gate}\mathcal{R}_{\text{gate}} + \lambda_{mask}\mathcal{R}_{\text{mask}}, \tag{20}$$

where $\Phi = \{\omega, W_g, W_n\}$, and $\lambda_{gate}$ and $\lambda_{mask}$ are balanced hyper-parameters. Recall the objective in Eq. 3, $\Gamma_{\Phi}$ can achieve better generalization due to the wider plane of the mixed distribution going over the unseen graphs.

## 4 EXPERIMENTS

In this section, we mainly focus on the graph classification tasks on the widely-used real-world datasets which encompass both observed (training) and unobserved (testing) data. Supplementary experiments (on the large-scale datasets / the node-level tasks) are provided in the Appendix C.4. Following common practice, we use the Accuracy (Acc) and Precision (Pre) on the OOD target dataset for measuring the generalization performance.

### 4.1 EXPERIMENT SETUP

**Datasets.** We evaluate our method on four datasets: MUTAG, PTC, REDDIT-B, and NCI1, following the same configurations as in (Xu et al., 2018). To simulate realistic domain shift scenarios, we partition each dataset based on the edge-to-node ratio, following established domain partitioning methods (Luo et al., 2024a; Zeng et al., 2024; Luo et al., 2024b; Wen et al., 2025; Wang et al., 2025). This strategy creates meaningful distributional differences between domains while maintaining the intrinsic properties of each dataset. Summary statistics of these datasets and detailed specifications of partitioning are provided in Appendix B.1. In this paper, we distinguish between domains using the notation "**A / B / T**". Specifically, "**A**" represents dataset slices with lower edge density, "**B**" refers to slices with intermediate edge density values, and "**T**" denotes the test set with higher edge density.

**Baselines.** First we pre-train models on each observable domain, resulting in multiple pre-trained models. Then, we perform graph models merging and evaluate the generalization performance on the unseen testing domain. We use three widely-adopted GNN architectures—GCN (Kipf & Welling, 2016), GAT (Veličković et al., 2017), and GIN (Xu et al., 2018)—as off-the-shelf models to be merged. Additionally, we use the form of (architecture-**A / B**) to distinguish GNNs trained on different source domains. For example, GCN-**A** refers to a GCN trained on source domain **A**. For ease of comparison, all GNNs used in our experiments are 2-layer networks with 32 feature dimensions. Since no known methods exist for merging GNN models with diverse architectures, we design two baseline approaches for reference: Inverse-X and Multi-GFKD. We compare our method with seven source-free graph domain generalization methods, which can be divided into three groups:

Table 1: Performance comparison across four datasets. The form (Architecture-A / B) indicates that this architecture is pre-trained on domain A/B. Highlighted are the top first, second results.

| Methods | REDDIT-B | | PTC | | MUTAG | | NCI1 | |
|---|---|---|---|---|---|---|---|---|
| | Acc/%↑ | Pre/%↑ | Acc/%↑ | Pre/%↑ | Acc/%↑ | Pre/%↑ | Acc/%↑ | Pre/%↑ |
| GCN-A | 25.03±6.67 | 35.55±32.38 | 48.97±3.59 | 50.38±4.31 | 31.25±8.00 | 37.47±31.87 | 49.62±6.53 | 57.85±3.97 |
| GAT-A | 24.21±10.20 | 27.05±34.91 | 48.10±3.67 | 53.30±8.56 | 26.88±0.94 | 16.22±18.79 | 49.91±3.66 | 58.06±1.17 |
| GIN-A | 22.46±8.28 | 18.47±20.35 | 47.93±4.42 | 50.98±7.43 | 28.91±6.02 | 27.81±31.88 | 52.58±1.98 | 60.46±1.52 |
| GCN-B | 66.10±3.59 | 60.64±4.96 | 49.90±2.85 | 49.77±10.69 | 32.03±9.79 | 33.86±33.23 | 61.61±3.22 | 62.42±2.53 |
| GAT-B | 61.78±26.86 | 55.36±26.77 | 49.38±4.16 | 49.84±10.85 | 27.81±2.86 | 19.94±26.10 | 60.75±1.89 | 63.14±1.43 |
| GIN-B | 58.80±18.56 | 56.74±7.01 | 50.12±5.59 | 55.86±8.87 | 42.03±12.49 | 49.24±26.51 | 65.02±1.66 | 66.79±1.11 |
| Avg-PTM | 52.47±5.46 | 51.66±8.29 | 50.20±1.95 | 51.73±3.43 | 31.48±2.88 | 32.42±12.33 | 56.58±2.12 | 61.45±1.13 |
| Ens-Prob | 33.65±25.66 | 36.12±39.06 | 50.17±2.58 | 56.64±6.99 | 29.84±5.65 | 35.86±35.35 | 58.05±4.36 | 62.65±1.51 |
| Ens-HighConf | 44.46±29.00 | 45.86±35.70 | 48.19±1.87 | 50.83±3.98 | 32.34±7.94 | 47.99±33.41 | 56.44±6.77 | 61.73±3.07 |
| Uni-Soup | 43.26±14.09 | 31.65±17.22 | 50.20±2.48 | 47.38±6.09 | 37.40±12.03 | 17.97±12.17 | 48.73±8.83 | 47.25±11.09 |
| Greedy-Soup | 47.35±8.89 | 50.70±9.62 | 50.17±2.50 | 42.75±9.43 | 31.46±6.23 | 13.91±8.51 | 38.64±10.43 | 28.67±11.81 |
| Inverse-X | 56.21±27.12 | 48.86±30.58 | 50.43±3.50 | 51.92±3.40 | 38.75±17.91 | 40.14±31.57 | 62.39±9.68 | 56.35±7.32 |
| Multi-GFKD | 54.35±11.40 | 37.96±11.40 | 50.77±1.3 | 44.43±4.39 | 44.36±8.42 | 29.74±10.37 | 47.57±4.84 | 36.75±7.42 |
| **OGMM** | **76.98±5.19** | **63.36±0.81** | **51.21±3.74** | **57.39±6.71** | **45.62±18.67** | **56.28±26.70** | **66.84±0.45** | **72.90±4.89** |

Table 2: Ablation study about different modules. Highlighted are the top first, second results.

| Variants | | REDDIT-B | | PTC | | MUTAG | | NCI1 | |
|---|---|---|---|---|---|---|---|---|---|
| | | Acc/% | Pre/% | Acc/% | Pre/% | Acc/% | Pre/% | Acc/% | Pre/% |
| Given Source | w/o Mask | 43.95±26.06 | 72.34±23.14 | 49.31±2.77 | 54.47±6.97 | 28.12±2.10 | 33.52±33.46 | 48.36±3.30 | 61.78±2.46 |
| | **OGMM** | **80.98±11.30** | **78.33±2.91** | **54.31±2.70** | **59.16±5.04** | **57.81±7.30** | **68.79±3.13** | **68.04±2.13** | **71.90±1.16** |
| Source Free | w/o MoE | 50.39±5.21 | 45.01±1.71 | 50.56±0.74 | 51.01±2.69 | 39.53±1.70 | 23.29±0.81 | 60.62±0.22 | 66.68±2.72 |
| | w/o Mask | 31.98±18.77 | 35.99±38.87 | 50.95±2.90 | 55.36±6.15 | 28.28±1.47 | 49.39±34.98 | 51.11±1.06 | 59.70±0.98 |
| | w/o $\mathcal{L}_{gen}$ | 41.15±26.95 | 39.75±36.20 | 48.88±5.23 | 48.61±5.63 | 45.31±22.96 | 25.81±22.96 | 52.69±11.61 | 55.82±3.46 |
| | **OGMM** | **76.98±5.19** | 63.36±0.81 | 51.21±3.74 | 57.39±6.71 | 45.62±18.67 | 56.28±26.70 | 66.84±0.45 | 72.90±4.89 |

- Ensemble learning methods include averaging the performance of the models (Avg-PTMs), averaging the output probabilities of the models (Ens-Prob), and selecting the prediction from the most confident model, defined as the one with the lowest entropy (Ens-HighConf).

- Model merging methods, include computing the element-wise mean of all models (Uni-Soup) (Choshen et al., 2022) and the selective merging approach (Greedy-Soup) (Wortsman et al., 2022).

- Generative methods include Inverse-X and Multi-GFKD. Inverse-X is a baseline variant of OGMM that uses random graph structures instead of our parameterized edge encoder. Multi-GFKD is an extension of GFKD (Deng & Zhang, 2021) to multi-teacher distillation.

## 4.2 EXPERIMENTAL RESULTS

**Main Results.** The comparisons of different models under the split-dataset scenarios are shown in Table 1. OGMM consistently outperforms individual pre-trained models across all datasets, demonstrating the MoE module's effectiveness in capturing distribution shifts and accurately allocating "sample-expert" pairs. Ensemble methods like Avg-PTMs, Ens-Prob, and Ens-HighConf show similar precision, suggesting that leveraging multiple models can improve generalization. In contrast, parameter merging methods (Uni-Soup, Greedy-Soup) perform poorly, highlighting that integrating model outputs is more effective for OOD problems. Compared to other source-free methods, OGMM sets a new state-of-the-art, achieving superior performance across datasets, especially on larger datasets like REDDIT-B and NCI1. While data generation-based methods (Inverse-X, Multi-GFKD) outperform fusion approaches, OGMM surpasses both, offering significant improvements. Unlike Inverse-X, which only learns node features, OGMM simultaneously learns node features and graph structures, enabling better recovery of domain-specific knowledge. Additionally, OGMM preserves more source-domain knowledge, maintaining the diversity of observable distributions.

**Analysis of Masks.** We apply masks to two parameter groups, MaskCL and MaskNN, across three GNN architectures to analyze mask placement impact. MaskCL applies masks to classifier parameters ($\theta_\Phi$) while freezing others; MaskNN applies masks to encoder parameters ($\theta_\Psi$). As shown in Figure 4, models fine-tuned exclusively on classifier parameters achieve competitive performance across datasets. The mask size accounts for only 20% of total parameters in a 2-layer GNN on average. This suggests domain-specific knowledge is concentrated in classifier parameters, making classifier fine-tuning more efficient. See Appendix C.1 for more results on other datasets. Additionally, we analyze

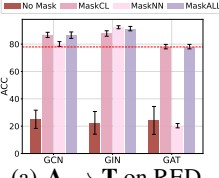 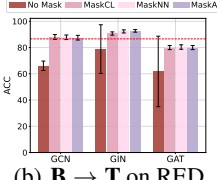 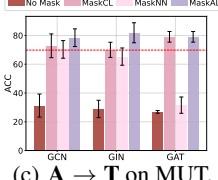 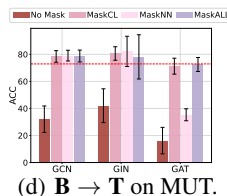

(a) **A** → **T** on RED.    (b) **B** → **T** on RED.    (c) **A** → **T** on MUT.    (d) **B** → **T** on MUT.

Figure 4: Impact of Mask Position on REDDIT-B (RED.) and MUTAG (MUT.). The form (**A** → **T**) means that a GNN pre-trained on domain **A** and fine-tuned on the **T**arget domain. The bar chart shows the model performance on the target domain, and the dashed line represents the average performance of masked models with different mask positions on this dataset.

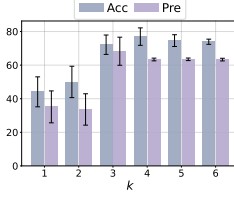 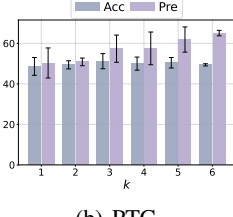 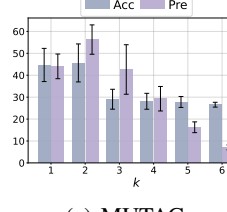 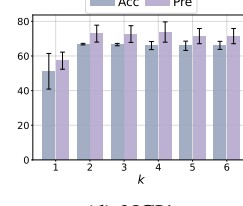

(a) REDDIT-B    (b) PTC    (c) MUTAG    (d) NCI1

Figure 5: The Effects of $k$ in $TopK$ Expert Selection on four datasets.

parameter changes after continuous fine-tuning across multiple domains. Results demonstrate that classifier parameters exhibit stabilizing characteristics after multiple fine-tuning rounds, providing evidence for our mask mechanism's effectiveness. The comprehensive parameter evolution analysis and associated visualizations are detailed in Appendix C.2.

**Ablation Studies.** To evaluate the efficacy of OGMM's components, we conduct an ablation study comparing five variant configurations, with comprehensive details provided in Appendix B.3 and quantitative results presented in Table 2. The variant "**OGMM (under Given Source condition)**", which leverages access to source domain data and incorporates additional trainable parameters, demonstrates superior performance as expected. The variant "**w/o Mask (under Given Source condition)**" only optimizes merging weights with fixed pre-trained parameters, performing similarly to Ens-Prob / Ens-HighConf from Table 1. Notably, our proposed OGMM achieves optimal results in the source-free setting, approaching the best performance despite the absence of source domain data, thus validating its capability for effective cross-domain knowledge transfer. Removing the MoE module, generator, or masks under the source-free constraint leads to performance declines, underscoring the critical contributions of these components.

**Impact of $TopK$ Expert Selection.** To investigate the effects of the hyper-parameter $k$ in the $TopK$ selector, we evaluate results across four datasets as shown in Figure 5. The performance changes reveal that selecting $k$ between 2 and 4 generally yields optimal results across all datasets. Most datasets exhibit similar trends, with accuracy improving as $k$ increases initially and then stabilizing at higher values. Notably, OGMM consistently outperforms the pre-trained baseline across most settings, confirming the effectiveness of our MoE module. In addition, these results show that the optimal choice requires dataset-specific tuning to accommodate varying dataset characteristics.

**Impact of the Number of Synthetic Samples.** Theoretically, OGMM can generate unlimited synthetic graphs for training, but their quality and diversity are limited by the pre-trained models, as noted by (Deng & Zhang, 2021). Figure 6 shows the relationship between OGMM's performance and the number of generated graphs on REDDIT-B and NCI1. OGMM achieves high performance even with a small fraction of synthetic graphs, as these effectively capture high-order domain knowledge, resulting in a concentrated and informative distribution.

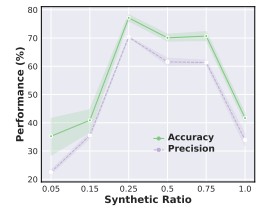 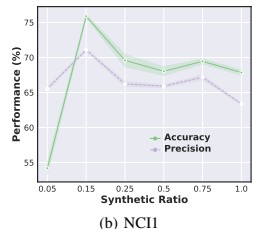

(a) REDDIT-B    (b) NCI1

Figure 6: Impact of the number of generative graphs. The horizontal axis is the ratio of generative samples to total source domain data.

**Manifold Visualization for Synthetic Graphs.** To further validate the effectiveness and diversity of the synthesized graphs, we visualize the data using t-SNE in Figure 7. The results present the class-wise distribution of real and synthetic data specifically on Domain A in NCI1. The visualization reveals that synthetic data aligns with the distribution of real data, indicating that our method successfully extracts domain-specific knowledge embedded in the pre-trained models.

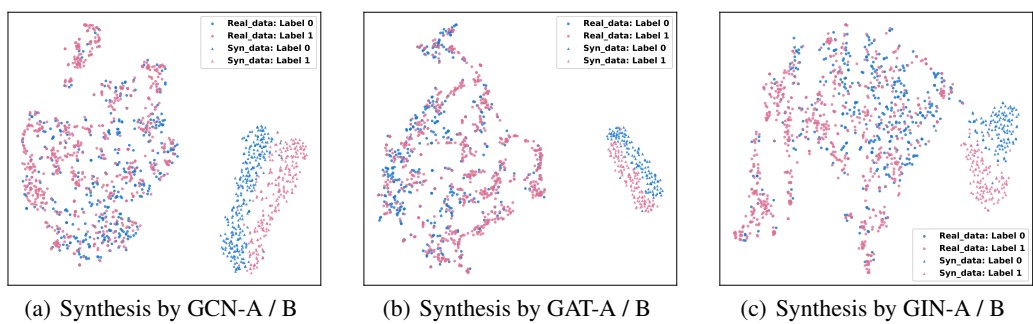

(a) Synthesis by GCN-A / B      (b) Synthesis by GAT-A / B      (c) Synthesis by GIN-A / B

Figure 7: The t-SNE Visualization of Real / Synthetic Samples with Label Distribution on Domain A.

**Analysis of Gate Distribution in OGMM.** To validate the rationality of our MoE-based merging design, we analyze the gating mechanism's behavior on both synthetic training data and real target domain data across four datasets, as shown in Figure 8. We compute the total weight assigned to each expert by aggregating gate assignments across all samples.

During the training phase with synthetic data (gray bars), the load distribution across experts remains relatively balanced, indicating that each expert receives approximately equal amounts of fine-tuning data. This balanced training ensures all experts are adequately optimized without bias toward any particular source domain. In contrast, during inference on real target domain data (purple bars), the gate distribution becomes significantly more discriminative. The gating mechanism effectively captures the distinguishable patterns learned by different experts and adaptively routes test samples to the most suitable expert based on distributional similarity. This shift in distribution reflects the varying relevance of different source domains to the target domain.

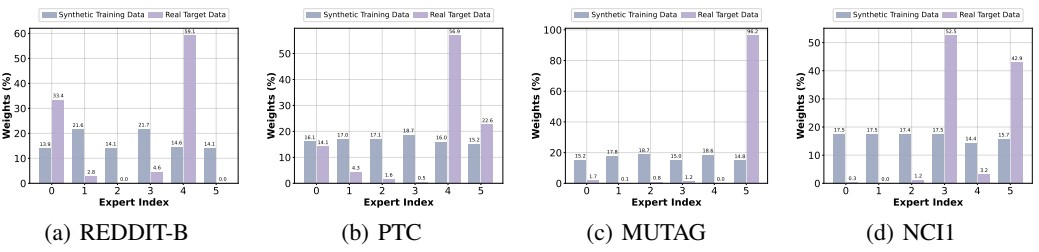

(a) REDDIT-B      (b) PTC      (c) MUTAG      (d) NCI1

Figure 8: Gates distribution on training data and real target data on four datasets.

## 5 CONCLUSION

This paper investigates the problem of Out-of-Distribution Graph Models Merging. The primary challenge lies in extracting knowledge from pre-trained GNNs and guiding their reuse to address the issue of model generalization. To tackle this challenge, we propose a novel out-of-distribution graph models merging framework. Our approach leverages graph generation and a fine-tuned MoE to adaptively optimize the model fusion process, enabling effective generalization under graph OOD scenarios. Extensive experiments on several real-world benchmarks confirm that the proposed approach outperforms state-of-the-art baselines.

## ACKNOWLEDGMENTS

The work of Ziyue Qiao was partially supported by the Guangdong Basic and Applied Basic Research Foundation (No. 2024A1515140114) and the National Natural Science Foundation of China (No. 62406056). The work of Pengyang Wang was funded by the Science and Technology Development Fund (FDCT), Macau SAR (File No. 001/2024/SKL).

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

This appendix contains details about mathematical proofs, experimental implementation, supplementary experiments, related works, limitations, and future works.

# A PROOF

## A.1 PROOF OF THEOREM 3.3

*Proof.* Given a domain $\mathcal{G}_i$ with two trained classifiers $f(\Theta_i)$ and $f(\Theta_j)$. These classifiers may have been trained on different source domains or under different conditions, leading to potentially divergent prediction behaviors. Based on (Ben-David et al., 2010), we can define the probability according to the distribution $\mathcal{G}_i$ that $f(\Theta_i)$ disagrees with $f(\Theta_j)$:

$$\mathcal{E}_i(f(\Theta_i), f(\Theta_j)) = E_{G \sim \mathcal{G}_i}[|f(\Theta_i, G) - f(\Theta_j, G)|]. \tag{21}$$

If the classifier $f(\Theta_i)$ is a good learner trained on $\mathcal{G}_i$, meaning it has achieved low training error and captures the underlying patterns of domain $\mathcal{G}_i$ effectively. We will find the generalization error of $f(\Theta_j)$ over $\mathcal{G}_i$:

$$\mathcal{E}_i(f(\Theta_i), f(\Theta_j)) = E_{G \sim \mathcal{G}_i}[|\mathcal{Y} - f(\Theta_j, G)|], \ \ \text{s.t.} f(\Theta_i, G) = \mathcal{Y}. \tag{22}$$

Following Eq. 21, Definition 3.2 can be formalized as follows:

$$d_{\mathcal{H}\Delta\mathcal{H}}(\mathcal{G}_i, \mathcal{G}_j) = 2 \sup_{f(\Theta_i), f(\Theta_j) \in \mathcal{H}} |\mathcal{E}_i(f(\Theta_i), f(\Theta_j)) - \mathcal{E}_j(f(\Theta_i), f(\Theta_j))|. \tag{23}$$

Substituting Eq. 22 into Eq. 23 yields:

$$\begin{aligned} d_{\mathcal{H}\Delta\mathcal{H}}(\mathcal{G}_i, \mathcal{G}_j) &= 2 \sup_{f(\Theta_i), f(\Theta_j) \in \mathcal{H}} |\mathcal{E}_i(\hat{y}_i, f(\Theta_j)) - \mathcal{E}_j(f(\Theta_i), \hat{y}_j)| \\ &\propto \sup_{f(\Theta_i), f(\Theta_j) \in \mathcal{H}} |\log p(\hat{y}_i | \mathcal{G}_i, f(\Theta_j)) - \log p(\hat{y}_j | \mathcal{G}_j, f(\Theta_i))| \\ \text{s.t.} \quad & f(\Theta_i, \mathcal{G}_i) = \hat{y}_i, f(\Theta_j, \mathcal{G}_j) = \hat{y}_j, \end{aligned} \tag{24}$$

which implies that the $\mathcal{H}\Delta\mathcal{H}$-Divergence of $\mathcal{G}_i$ and $\mathcal{G}_j$ depends on the cross-validation results of the respective optimized classifiers. Note that the disparity difference function represented by Eq. 24 is symmetric and obeys the triangle inequality. So we can build a cross-domain objective function based on a set of pre-trained models:

$$\begin{aligned} \arg \min_{\omega, \mathcal{G}} & \sum_{i,j}^{M} d_{\mathcal{H}\Delta\mathcal{H}}(\mathcal{G}_i, \mathcal{G}_j) \\ &\propto \arg \min_{\omega, \mathcal{G}} \sum_{i,j}^{M} |\log p(\hat{y}_i | \mathcal{G}_i, f(\Theta_j), \omega^j) - \log p(\hat{y}_j | \mathcal{G}_j, f(\Theta_i), \omega^i)| \\ \text{s.t.} \quad & \mathcal{G}_i = \arg \max_{\mathcal{G}} \log p(\hat{y}_i | \mathcal{G}, f(\Theta_i)), \forall \mathcal{G}_i \in \mathcal{G}, \end{aligned} \tag{25}$$

where $\omega^i$ is a learnable parameter for $f(\Theta_i)$. Optimization 25 achieves two purposes: (1) sufficient extraction of knowledge from the models to compose a more generalized mixture of distributions of the data $\mathcal{G}$, and (2) optimization of the added parameters to fine-tune the individual model on the mixture of distributions. Details on solving the Optimization 25 can be found in A.2.

Due to the rule of linear combination, $\Gamma = \sum_i^M \alpha_i f(\Theta_i)$, we have $\Gamma \in \mathcal{H}$. Thus, the optimization objective of Optimization 25 is to identify an appropriate discriminative function $\Gamma$ that minimizes the generalization error across arbitrary marginal distributions. Therefore, the upper bound of the generalization error for $\Gamma$ on the target domain (the mixture distribution according to Assumption 3.1) depends on the sum of the cross-validation errors of sub-learners. □

## A.2 DETAILS ON SOLVING OPTIMIZATION 25

Optimization 25 is a non-convex problem, which is hard to solve. We relax it via the triangle inequality. At the same time, we replace the log-likelihood function with regular cross-entropy loss, and finally get a two-stage target function:

$$\arg\min_{\omega, \mathcal{G}^*} \sum_{i,j}^{M} \mathcal{C}_{G \in \mathcal{G}_i^*}(\hat{y}_i, f(\Theta_j, \omega^j, G)) \quad [Sec.\ 3.3]$$

$$\text{s.t.} \mathcal{G}_i^* = \arg\min_{\mathcal{G}} \mathcal{C}_{G \in \mathcal{G}}(\hat{y}_i, f(\Theta_i, G)), \forall \mathcal{G}_i^* \in \mathcal{G}^* \quad [Sec.\ 3.2], \tag{26}$$

where $\mathcal{C}_{G \in \mathcal{G}}(\cdot)$ denotes the cross-entropy loss function on distribution $\mathcal{G}$. $\mathcal{G}_i^* \in \mathcal{G}^*$ is a batch of generated samples. The ideal is for any fine-tuned model to have a small a posteriori error on any sampled data belonging to $\mathcal{G}^*$, which is very difficult to achieve. In practice, we simply approximate it using a finite number of samples. Meanwhile, we substitute $\Gamma = \sum_i^M \alpha_i f(\Theta_i)$ into Optimization 26:

$$\arg\min_{\omega, \alpha} \sum_i^N \mathcal{C}_{G_i \in \mathcal{G}^*}(\hat{y}_i, \sum_j^M \alpha_j \cdot f(\Theta_j, \omega^j, G_i)), \tag{27}$$

where $N$ and $M$ denote the number for samples and models. $\hat{y}_i$ is the label of $G_i$, and $\alpha_i$ is the fusion weights, combining different models for different samples.

According to Theorem 4 in (Ben-David et al., 2010), for any $\delta \in (0, 1)$, with probability at least $(1 - \delta)$, the error bound of the merged function $\Gamma$ on the target domain $\mathcal{G}_T$ can be defined as follows:

$$\epsilon_T(\Gamma) \leq \epsilon_T(h_T^*) + \sum_{j=1}^{M} \alpha_j \left(2\lambda_j + d_{\mathcal{H}\Delta\mathcal{H}}(\mathcal{G}_j, \mathcal{G}_T)\right)$$

$$+ 4\sqrt{\left(\sum_{j=1}^{M} \frac{\alpha_j^2}{\beta_j}\right) \left(\frac{2d \log(2(N+1)) + \log\left(\frac{4}{\delta}\right)}{N}\right)}, \tag{28}$$

where $\mathcal{H}$ is a hypothesis space of VC dimension $d$. $h_T^* = \min_{h \in \mathcal{H}} \epsilon_T(h)$ is the target error minimizer. $N$ represents the sum of the number of all samples in all source domains, and $\beta_j = \frac{N_j}{N}$ is the ratio of the samples from the $j$-th domain. $\alpha$ is a fixed weight vector. $\lambda_j = \min_{h \in \mathcal{H}}\{\epsilon_T(h) + \epsilon_j(h)\}$ means the optimal cross-domain generalization error (defined in $\mathcal{H}$), and this term corresponds to our expectations for the fine-tuned pre-trained models.

## A.3 PROOF OF THE EXPANSION OF $\epsilon_i(f(\Theta_i))$

*Proof.* First we provide the definition of $\mathcal{H}$-Divergence between $\mathcal{G}_i$ and $\mathcal{G}_j$:

$$d_{\mathcal{H}}(\mathcal{G}_i, \mathcal{G}_j) = 2 \sup_{f(\Theta_i), f(\Theta_j) \in \mathcal{H}} |Pr_{G \sim \mathcal{G}_i} f(\Theta_i, G) - Pr_{G \sim \mathcal{G}_j} f(\Theta_j, G)|, \tag{29}$$

where $Pr_{G \sim \mathcal{G}_i} f(\Theta_i, G)$ means the prediction of $f(\cdot)$ on $\mathcal{G}_i$. Suppose that $\mathcal{G}^*$ is the mixed distribution of the set $\{\mathcal{G}_i\}_{i=1}^{M}$, and $\mathcal{G}^*$ can be defined as follows:

$$\mathcal{G}^* = \sum_{i=1}^{M} \alpha_i \mathcal{G}_i, \tag{30}$$

where $\alpha_i$ is the mixing coefficient. Note that when no training data is available but model parameters are known, we can optimize the inputs by minimizing the empirical error $\epsilon_i(f(\Theta_i))$ on $\mathcal{G}_i$ (i.e., $argmax_{\mathcal{G}_i^*} \log p(\hat{y}_i|\mathcal{G}_i^*, f(\Theta_i))$) to generate data (Deng & Zhang, 2021). This process can be seen as narrowing the $\mathcal{H}$-Divergence between $\mathcal{G}_i^*$ and $\mathcal{G}_j^*$ according to Eq. 29. When $Pr_{G \sim \mathcal{G}_i} f(\Theta_i, G)$ is close enough to $\hat{y}_i$ ($f(\Theta_i)$ fits well enough on $\mathcal{G}_i$), we can assume that $\mathcal{G}_i^*$ samples from $\mathcal{G}^*$, which still has the $\mathcal{H}\Delta\mathcal{H}$-Divergence:

$$d_{\mathcal{H}\Delta\mathcal{H}}(\mathcal{G}_i^*, \mathcal{G}_j^*) = 2 \sup_{h_i, h_j \in \mathcal{H}} |\mathcal{E}_i(\hat{y}_i, h_j) - \mathcal{E}_j(h_i, \hat{y}_j)|$$

$$\propto \sup_{h_i, h_j \in \mathcal{H}} |\log p(\hat{y}_i|\mathcal{G}_i^*, h_j) - \log p(\hat{y}_j|\mathcal{G}_j^*, h_i)|$$

$$\text{s.t.} \quad \mathcal{G}_i^* = \arg\max_{\mathcal{G}} \log p(\hat{y}_i|\mathcal{G}, f(\Theta_i)), \forall \mathcal{G}_i^* \in \mathcal{G}^*, \tag{31}$$

where $(h_i, h_j)$ is a set of functions used to define the lower bound of $d_{\mathcal{H}\Delta\mathcal{H}}(\mathcal{G}_i^*, \mathcal{G}_j^*)$. We can use the fine-tunable model $f(\Theta_i)$ and $f(\Theta_j)$ to approximate $h_i$ and $h_j$ with added parameters $\omega$. □

## B IMPLEMENTATION DETAILS

### B.1 DATASETS DETAILS

Table 3 shows the summary statistics of the datasets used in Sec. 4.2.

Table 3: Summary of datasets.

|               | MUTAG | PTC    | REDDIT-B  | NCI1    |
|---------------|-------|--------|-----------|---------|
| #Graphs       | 188   | 344    | 2,000     | 4,110   |
| #Classes      | 2     | 2      | 2         | 2       |
| #Feature Dim  | 7     | 19     | 37        | 1       |
| #Nodes        | 3,371 | 8,792  | 859,254   | 122,747 |
| #Edges        | 7,442 | 17,862 | 1,991,016 | 265,506 |
| Avg #Nodes    | 17.93 | 14.29  | 429.62    | 29.87   |
| Avg #Edges    | 39.59 | 51.92  | 995.51    | 64.60   |

To simulate realistic out-of-distribution scenarios, we partition each graph-level dataset into multiple domains based on graph edge density, following established domain adaptation methods (Luo et al., 2024a; Zeng et al., 2024; Luo et al., 2024b; Wen et al., 2025; Wang et al., 2025). Edge density serves as a fundamental structural characteristic that creates meaningful distributional shifts across graph domains (Fu et al., 2024). For each graph-level dataset, we calculate the edge density $\rho = \frac{2|E|}{|V|(|V|-1)}$ for every graph and partition the data into domains based on density value, ensuring that each domain contains graphs with similar structural complexity while maintaining sufficient distributional differences between domains. The complete partitioning implementation is available in our code repository. This partitioning strategy reflects real-world scenarios where models encounter graph structural variations, such as in molecular datasets like MUTAG, where density variations correspond to distinct chemical families: dense graphs typically represent highly conjugated aromatic systems with extensive double bond networks, while sparse graphs correspond to simpler aliphatic structures that exhibit different toxicity mechanisms and create natural domain boundaries.

Similar density-driven domain shifts appear across various graph learning applications, from social networks where active users generate dense interaction patterns while inactive users create sparse connectivity, to spatiotemporal trajectory analysis where dense urban movement patterns differ significantly from sparse rural trajectories. These distribution shifts lead to model performance degradation, which is precisely the generalization challenge our method aims to address.

### B.2 PARAMETERS SETTING

In our experimental setup, the number of generated samples $N$ and the number of source domains $M$ are known a priori, not tunable hyperparameters. For the fake graphs generation stage, the number of epochs is set to 200. $\tau = 0.2$ in Eq. 8 controls sampling stability. For the model merging stage, the number of epochs is set to 20. The AdamW optimizer (Shazeer & Stern, 2018) is used for gradient descent. $\gamma_v = \gamma_p = 0.9$ in Eq. 19 control parameter changes from pretrained models to preserve knowledge. The hyper-parameters $\lambda_{gate}$ and $\lambda_{mask}$ in the merging function, i.e., Eq. 20, are chosen from $\{10^{-2}, 10^{-1}, 1, 10, 100\}$, and the value of $k$ for the TopKSelector, i.e., Eq. 15, is chosen from $\{1, 2, 3, 4, 5\}$. We report the mean results and standard deviations of ten runs.

### B.3 ABLATION VARIANTS

We evaluate six configurations to analyze the contribution of each component:

**Given Source variants:** (1) Variant "**w/o Mask**" removes the parameter masks from pre-trained model classifiers while maintaining access to source data. (2) Variant "**OGMM**" represents our full method with access to source domain data, serving as an upper bound for performance.

**Source Free variants:** (3) Variant "**w/o MoE**" eliminates the MoE module and uses simple averaging of masked GNN predictions. (4) Variant "**w/o Mask**" removes the masks added to the classifiers of each pre-trained model in the source-free setting. (5) Variant "**w/o $\mathcal{L}_{gen}$**" removes the graph generation objective (Eq. 11), training the model merging stage with randomly generated noise graphs instead of our synthesized graphs. (6) Variant "**OGMM**" represents our full method under the source-free setting.

## B.4 ALGORITHM ANALYSIS

We analyze the computational complexity of our model to show its efficiency. Let $|V|$ denote the total number of generated nodes, $|E|$ represent the number of generated edges, $d_{in}$ and $d_{mid}$ indicate the dimensions of the initial and intermediate layer features, respectively. The computational complexity of the model during the fake graphs generation stage is given by: $O(|V|^3(d_{in} \cdot d_{mid} + d_{mid}^2) + (|E| \cdot d_{mid} + |V| \cdot d_{mid}^2))$. Generally speaking, $d_{in}$ and $d_{mid}$ are significantly smaller than $|V|$ or $|E|$. So the time complexity of the first stage of OGMM is $O(|V|^3 + |E| + |V|)$. The computational complexity of the second stage is $O(d_{mid} \cdot m + m(|E| \cdot d_{mid} + |W| \cdot |V| \cdot d_{mid}^2))$, where $m$ denotes the number of pre-trained models and $|W|$ represents the scale of masks. Therefore, the time complexity of OGMM is $O(|V|^3 + |E| + |V| + m(|E| + |W| \cdot |V|))$. Although our model demonstrates effectiveness, it has comparable complexity with the existing baselines.

The algorithm is shown in Algorithm 1. During the experiments, we use one NVIDIA GeForce RTX 4090D GPU to train and perform inference.

---

**Algorithm 1** Procedure of OGMM

---

**Input:** pre-trained graph models $\{f(\Theta_i)\}_{i=1}^{M}$.
**Output:** The predicting labels on the target samples.
  // First stage: Graphs Generation
  **for** $i = 1$ **to** $M$ **do**
    Initialize the domain-specific graph features $\{X^i | X^i \sim \mathcal{N}(0, I)\}_{i=1}^{M}$ and arbitrary labels;
    **while** not converged **do**
      Generate graph structures $\{A^i\}_{i=1}^{M}$ by generators $\{\mathcal{P}_i\}_{i=1}^{M}$ (Eq. 8);
      Update $\{\mathcal{P}_i\}_{i=1}^{M}$ and $\{A^i\}_{i=1}^{M}$ by minimizing the generation loss $\mathcal{L}_{gen}$ (Eq. 11);
    **end while**
  **end for**
  // Second stage: Graph Models Merging
  Concatenate the generative datasets into $\mathcal{G}^*$;
  Initialize the fine-tuning masks $\omega$ in Eq. 13 and the gating layer in Eq. 15;
  **while** not converged **do**
    Update $\{\omega, W_g, W_n\}$ by minimizing the merging loss $\mathcal{L}_{merge}$ (Eq. 20).
  **end while**

---

Besides, we further provide the running time comparison between OGMM and SOTA baselines in generating, training and testing phases in Tables 4–6 to verify effectiveness of the proposed method.

Table 4: Generating Time (in seconds) Comparison on four datasets.

| Methods/Datasets | REDDIT-B | PTC | MUTAG | NCI1 |
|---|---|---|---|---|
| Inverse-X | 2030.95 | 88.07 | 63.17 | 452.02 |
| Multi-GFKD | 4052.60 | 1640.38 | 743.02 | 2043.15 |
| OGMM | 3448.28 | 134.17 | 126.69 | 827.76 |

For generation time, OGMM is significantly faster than Multi-GFKD (1.2-15× speedup across datasets). Compared to Inverse-X, which is a simplified version of OGMM without graph structure optimization, OGMM requires additional time but delivers better performance. In the training phase, OGMM achieves the best efficiency, being 2-4× faster than traditional GNN methods and outperforming Multi-GFKD. Regarding testing time, OGMM is slightly slower than baselines but still operates at millisecond to second scale, which is acceptable for practical applications. Overall, OGMM shows good computational efficiency, especially in the training phase.

Table 5: Training runtime (in seconds) comparison on four datasets, the results are recorded at the time of running 100 epochs for fairness.

| Methods/Datasets | REDDIT-B | PTC | MUTAG | NCI1 |
|---|---|---|---|---|
| GCN | 97.55 | 17.50 | 9.31 | 116.89 |
| GIN | 112.42 | 19.09 | 9.47 | 175.76 |
| GAT | 144.30 | 14.01 | 25.26 | 206.27 |
| Multi-GFKD | 51.66 | 49.02 | 30.54 | 51.06 |
| OGMM | 41.28 | 8.25 | 8.92 | 45.99 |

Table 6: Testing runtime (in seconds) comparison on four datasets.

| Methods/Datasets | REDDIT-B | PTC | MUTAG | NCI1 |
|---|---|---|---|---|
| GCN | 0.04 | 0.01 | 0.01 | 0.07 |
| GIN | 0.04 | 0.01 | 0.01 | 0.09 |
| GAT | 0.06 | 0.01 | 0.01 | 0.08 |
| Uni-Soup | 0.05 | 0.03 | 0.03 | 0.09 |
| Greedy-Soup | 0.07 | 0.03 | 0.02 | 0.11 |
| Multi-GFKD | 0.09 | 0.02 | 0.01 | 0.16 |
| OGMM | 0.10 | 0.07 | 0.05 | 0.25 |

## C  SUPPLEMENTARY EXPERIMENTS

### C.1  POSITION OF MASKS

We conducted additional experiments on PTC and NCI1 to analyze the impact of mask placement. As shown in Figure 9, the results on these datasets further validate the conclusion: incorporating masks into the classifier (MaskCL) achieves performance comparable to the average of the three mask-tuning methods.

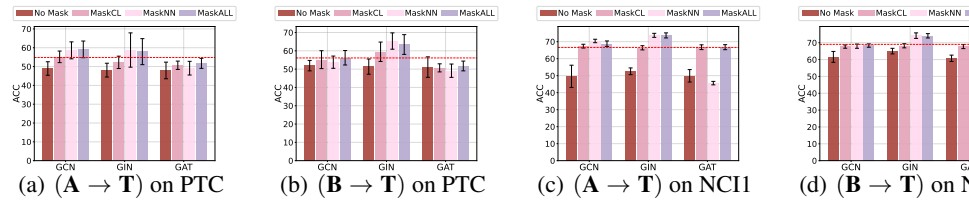

(a) $(\mathbf{A} \to \mathbf{T})$ on PTC  (b) $(\mathbf{B} \to \mathbf{T})$ on PTC  (c) $(\mathbf{A} \to \mathbf{T})$ on NCI1  (d) $(\mathbf{B} \to \mathbf{T})$ on NCI1

Figure 9: Impact of Mask Position on PTC and NCI1. The form $(\mathbf{A} \to \mathbf{T})$ means that a GNN pre-trained on domain $\mathbf{A}$ and fine-tuned on the $\mathbf{T}$arget domain. The bar chart shows the model performance on the target domain, and the dashed line represents the average performance of masked models with different mask positions on this dataset.

### C.2  GRADIENT TRENDS

Following the criteria outlined in (Wang et al., 2022), we calculate the proportion of large gradients for each model layer, as illustrated in Figures 10 - 12. The selected criterion is the number of parameters with a variation magnitude exceeding 0.001. Specifically, we partition the dataset into more subsets (10 / 20) based on edge density, and sequentially use these subsets to continuously train the selected GNNs. Then we use the selected criterion to analyze the degree of parameter variation within each module. Notably, the classifier weights exhibit a more pronounced decreasing trend in parameter changes compared to other layers. This further indicates that fine-tuning the classifier parameters facilitates the model's ability to learn invariant representations, thereby enhancing the

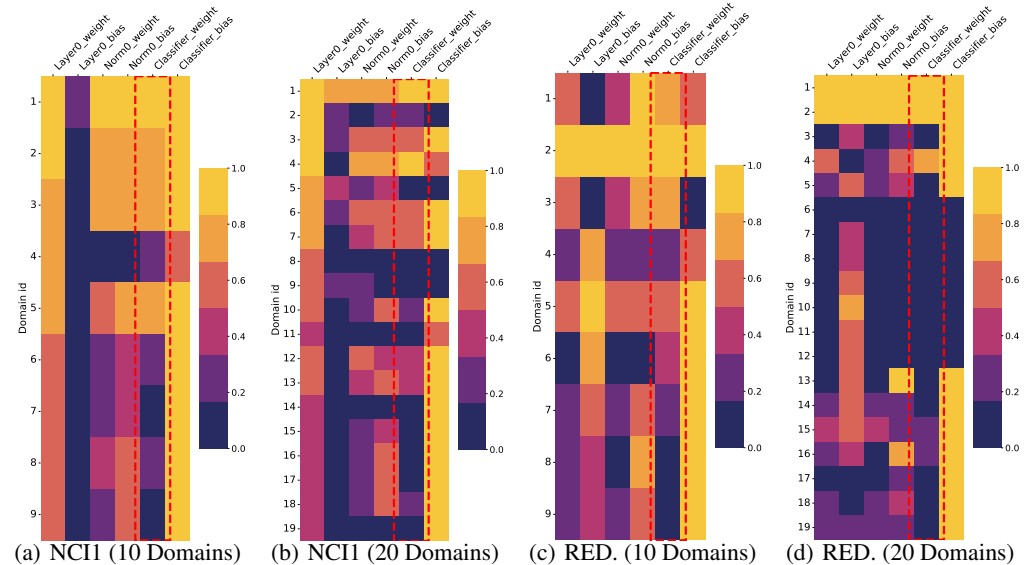

Figure 10: Gradient trends in 2-layer GCN on NCI1 and REDDIT-B. (RED). It illustrates the evolution of parameters at each layer of GCN as the model is trained with an increasing number of domain data, based on a specified criterion.

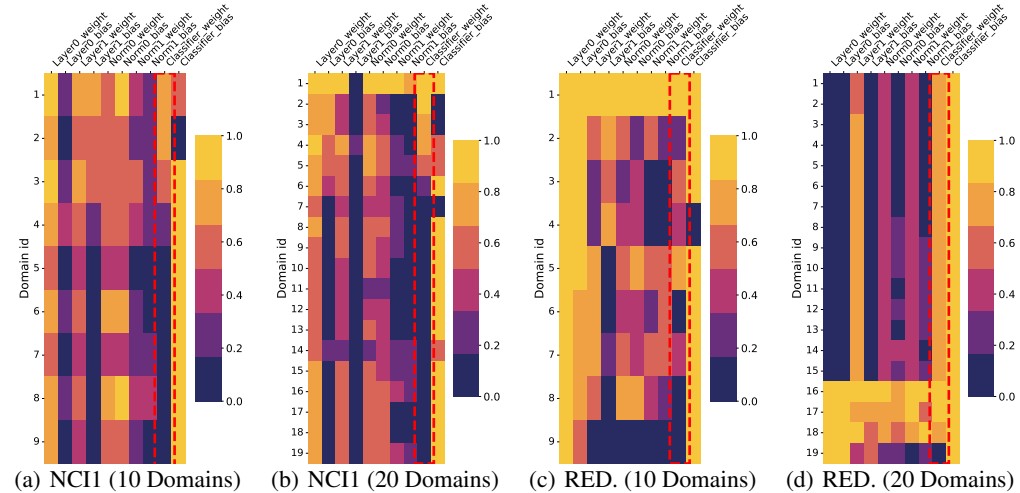

Figure 11: Gradient trends in 2-layer GIN on NCI1 and REDDIT-B (RED.). It illustrates the evolution of parameters at each layer of the GIN as the model is trained with an increasing number of domain data, based on a specified criterion.

generalization capability of individual sub-learners. The results confirm the insight provided in Sec. 4.2: the classifier weights gradually stabilize after training on multiple domains, indicating that these parameters capture cross-domain invariant knowledge.

## C.3 PERFORMANCE OF HOMOGENEOUS BACKBONES MERGING

Leveraging the MoE architecture, OGMM imposes no explicit constraints on the underlying model architectures. To further assess the generalizability of OGMM within homogeneous GNN backbones, we reduce the number of pre-trained models and employ only the widely adopted GCN for fusion. As presented in Table 7, even with the integration of just two basic GCNs, OGMM outperforms

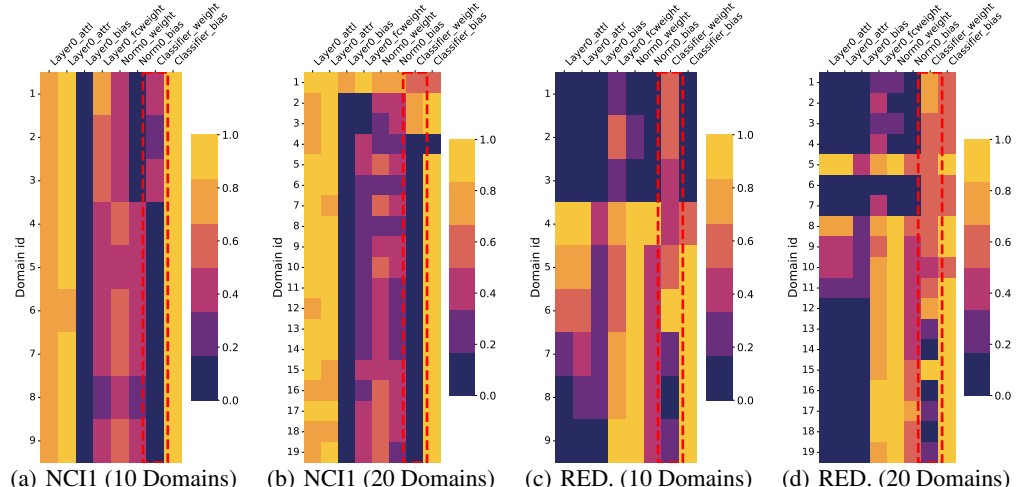

Figure 12: Gradient trends in 2-layer GAT on NCI1 and REDDIT-B (RED.). It illustrates the evolution of parameters at each layer of the GAT as the model is trained with an increasing number of domain data, based on a specified criterion.

existing fusion methods, setting a new state-of-the-art across all four datasets. Notably, compared to the results in Table 1, most performance metrics for OGMM improve upon reducing the number of pre-trained models. This suggests that, while the total knowledge volume remains constant, increased diversity among experts leads to the introduction of additional errors.

Table 7: Data performance comparison across four datasets. The experimental setup was identical to Table 1, except that the base models used only two GCNs trained from different domains. Highlighted are the top first, second results.

| Methods | REDDIT-B | | PTC | | MUTAG | | NCI1 | |
|---|---|---|---|---|---|---|---|---|
| | ACC/%↑ | Pre/%↑ | ACC/%↑ | Pre/%↑ | ACC/%↑ | Pre/%↑ | ACC/%↑ | Pre/%↑ |
| Avg-PTM | 53.85±3.74 | 59.79±15.42 | 50.43±2.28 | 51.08±5.14 | 31.64±8.26 | 35.67±26.85 | 55.62±4.52 | 60.14±3.13 |
| Ens-Prob | 42.25±24.82 | 48.29±36.58 | 51.72±2.94 | 55.27±3.90 | 28.12±2.42 | 34.08±33.79 | 55.09±6.26 | 60.62±2.61 |
| Ens-HighConf | 50.16±27.47 | 54.6±34.05 | 52.76±4.25 | 56.87±5.44 | 29.84±7.86 | 28.38±32.63 | 54.87±6.86 | 60.15±3.65 |
| Uni-Soup | 44.46±28.49 | 43.62±33.01 | 49.83±4.25 | 44.27±15.4 | 35.62±18.14 | 16.37±18.62 | 50.88±12.11 | 56.72±7.42 |
| Greedy-Soup | 50.06±29.28 | 57.41±28.04 | 48.71±2.99 | 39.76±14.58 | 36.56±15.66 | 22.94±24.54 | 49.12±15.37 | 54.57±17.81 |
| Inverse-X | 64.49±23.37 | 70.15±12.31 | 53.53±1.47 | 51.16±3.27 | 37.5±12.58 | 40.41±29.02 | 65.65±0.67 | 54.00±8.31 |
| Multi-GFKD | 64.10±36.88 | 64.03±36.75 | 54.17±22.91 | 38.87±17.39 | 46.56±12.53 | 34.94±27.74 | 38.02±15.49 | 54.88±21.28 |
| **OGMM** | **80.64±1.71** | **76.81±9.16** | **55.34±0.34** | **59.87±2.60** | **50.78±20.40** | **43.58±25.35** | **66.38±0.04** | **63.12±2.76** |

## C.4 PERFORMANCE ON LARGE-SCALE DATASETS

Table 8: Additional experiments on five large-scale datasets. The experimental setup was identical to Table 1. Highlighted are the top first, second results.

| Task | Graph Classification | Node Classification | | | |
|---|---|---|---|---|---|
| Dataset | ogbg-Molhiv | ogbn-Arxiv | Twitch | Facebook-100 | Elliptic |
| | Acc/%↑ | Acc/%↑ | Acc/%↑ | Acc/%↑ | Acc/%↑ |
| Avg-PTM | 93.40±0.14 | 48.03±0.34 | 45.43±1.32 | 51.45±0.47 | 60.31±1.45 |
| Ens-Prob | 95.46±0.18 | 46.12±0.32 | 48.41±2.29 | 46.82±0.07 | 77.81±0.98 |
| Ens-HighConf | 96.52±0.09 | 44.61±1.38 | 47.89±2.48 | 46.66±0.09 | 81.70±5.54 |
| Uni-Soup | 94.69±0.03 | 25.63±2.90 | 55.80±3.38 | 55.04±3.75 | 82.39±0.42 |
| Greedy-Soup | 78.61±38.16 | 32.63±3.75 | 53.29±3.36 | 55.51±1.46 | 82.08±1.30 |
| Inverse-X | 56.06±11.5 | 38.70±18.42 | 52.73±1.46 | 53.47±4.78 | 82.79±0.16 |
| Multi-GFKD | 70.43±36.68 | 15.00±0.54 | 51.33±1.99 | 54.23±1.52 | 82.65±0.21 |
| **OGMM** | **96.72±0.91** | **53.38±0.01** | **59.45±0.85** | **56.89±0.05** | **82.89±0.09** |

We extend our method to larger datasets for validation. We conduct graph-level classification tasks on ogbg-Molhiv (Hu et al., 2020), which contains 41,127 molecular graphs where each graph represents a chemical compound. Similar to our approach in Sec. 4.2, we use edge density as the criterion for domain partitioning, maintaining consistency with the partitioning method and pre-trained models described previously. For node-level classification tasks, we evaluate our approach on ogbn-Arxiv (Hu et al., 2020), Twitch (Rozemberczki et al., 2021), Facebook-100 (Traud et al., 2012) and Elliptic (Pareja et al., 2020) datasets. The ogbn-Arxiv dataset contains 169,343 nodes representing papers from 40 subject areas. We follow the domain partitioning method (Qiao et al., 2025) based on the temporal shifts and partition the pre-2017 data into two domains (1971-2013 and 2014-2017) for pre-training GNNs, and use the 2018-2020 data to test our model. The Twitch dataset contains 36,890 nodes representing users across seven regional networks. We pre-train GNNs using two groups of regions: (DE, ENGB, ES) and (FR, PTBR, RU), and evaluate OGMM on the TW region. The Facebook-100 consists of multiple social networks from different regions. The Elliptic is a Bitcoin transactions network dataset, includes graphs from different time steps.

As shown in Table 8, OGMM consistently establishes new state-of-the-art results across all benchmarks, significantly outperforming existing fusion methods. While traditional generative methods (e.g., Inverse-X and Multi-GFKD) struggle on large-scale datasets such as ogbg-Molhiv and ogbn-Arxiv often underperforming even simple ensemble baselines, OGMM maintains a clear advantage. This can be attributed to its generator design, which enables more stable and expressive expert modeling. In particular, compared with Inverse-X, OGMM yields substantial improvements across all datasets, highlighting its robustness to scale and task variation. The consistent superiority of OGMM underscores its effectiveness in integrating diverse knowledge sources while mitigating the instability typically introduced by generative fusion under large data regimes.

## C.5 PARAMETERS ANALYSIS

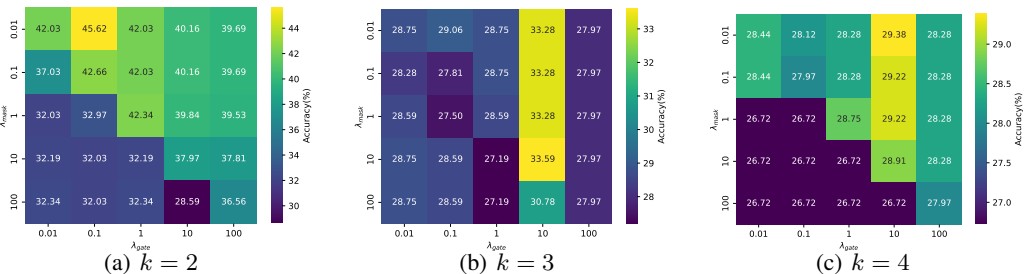

Figure 13: Hyper-parameter sensitivity for OGMM on MUTAG.

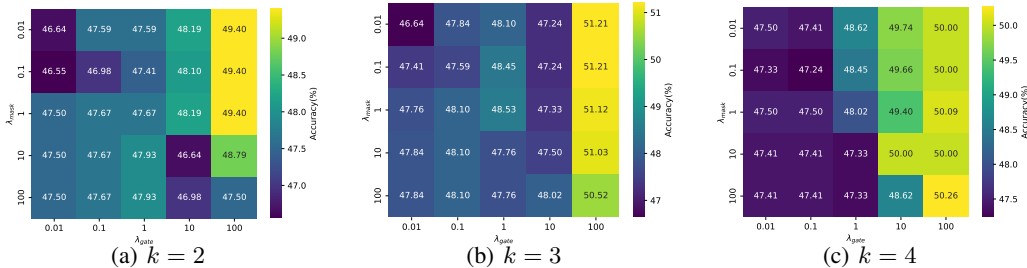

Figure 14: Hyper-parameter sensitivity for OGMM on PTC.

**Discussion about the Hyper-parameters.** We conducted extensive experiments on four datasets mentioned in Sec. 4.2 to analyze the impact of hyper-parameters $\{k, \lambda_{gate}, \lambda_{mask}\}$ on model performance, as shown in Figures 13 - 16 . Notably, on small datasets like MUTAG and PTC, the influences of $\{k, \lambda_{gate}\}$ are more pronounced due to the larger variations in pre-trained models caused by limited data. In this case, the fusion process has a more significant effect on the results. On

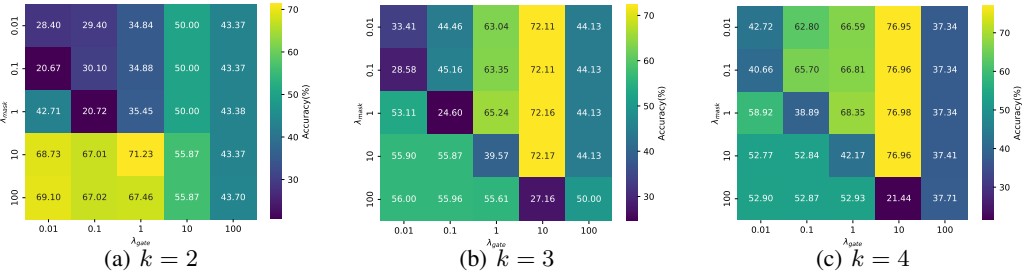

Figure 15: Hyper-parameter sensitivity for OGMM on REDDIT-B.

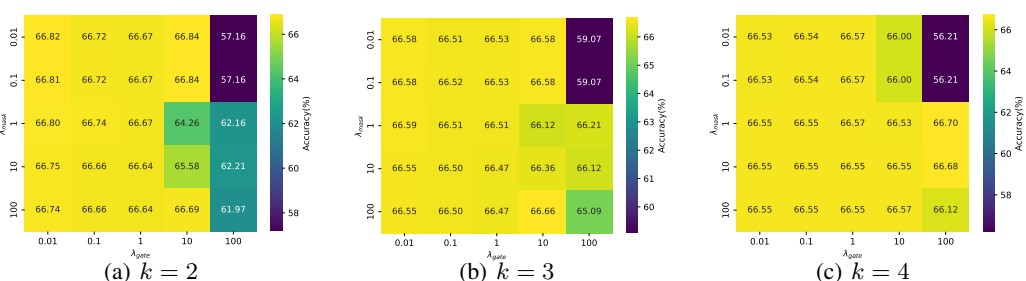

Figure 16: Hyper-parameter sensitivity for OGMM on NCI1.

larger datasets such as REDDIT-B and NCI1, $\lambda_{mask}$ plays a more crucial role, with the fine-tuning process ultimately determining the performance ceiling of the merged model.

**Discussion about the Number of Domains.** According to Optimization 12, OGMM can integrate multiple pre-trained models from different domains, with generalization improving as domain diversity grows. However, constructing datasets divided into infinite domains is impractical. Consequently, experiments rely on datasets with limited samples, where increasing the number of manually defined domains reduces the sample size per domain, impacting pre-trained model quality. This explains the trend in Figure 17, where OGMM's performance declines, and error rates rise as the number of domains increases.

### C.6 VISUALIZATION

We visualize the real and generative graphs obtained from MUTAG and NCI1, as shown in Figures 18 - 19. The visual comparison reveals some similarities between the graphs learned by OGMM and the real graphs, highlighting the model's ability to capture meaningful domain knowledge.

## D RELATED WORKS

### D.1 GRAPH DOMAIN GENERALIZATION

A growing body of research on Graph Domain Generalization has garnered increasing attention in recent years. Approaches such as (Qiao et al., 2023; Sun et al., 2024; Chen et al., 2024b; Yuan et al., 2024) concentrate on learning representations that remain stable and invariant across diverse environments. In parallel, methods like (Sui et al., 2022; Chen et al., 2024a; Gui et al., 2024; Fan et al., 2023) employ a causal inference framework to uncover relationships between data and labels that are robust to distribution shifts. Other techniques, including (Lu et al., 2024; Li et al., 2023; Jia et al., 2024), focus on improving model generalization by employing data augmentation strategies. Regardless of architectural differences, the effectiveness of these learning strategies is largely contingent on the precise acquisition, partitioning, and labeling of training data. Notably, the majority of existing approaches necessitate access to datasets with clearly delineated data from

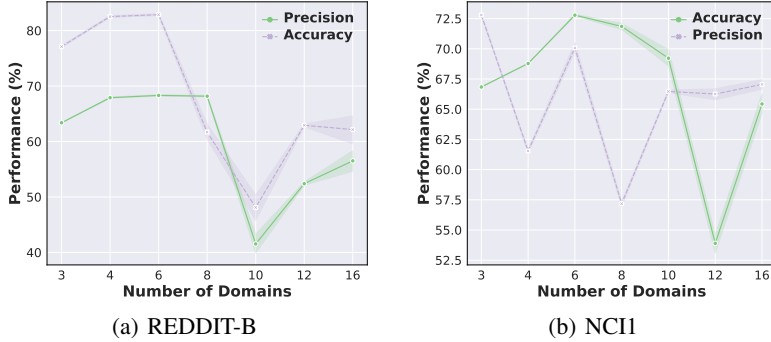

(a) REDDIT-B  (b) NCI1

Figure 17: Ablation studies regarding the number of domains. The horizontal axis indicates the number of source domains.

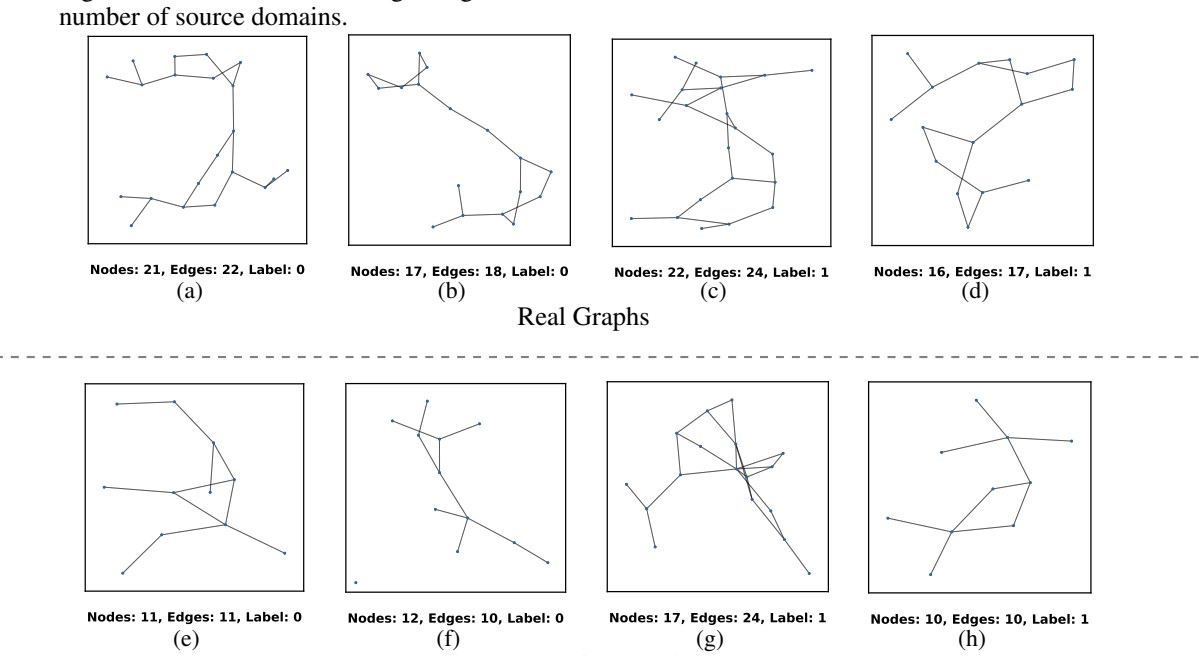

Figure 18: Graph visualization on MUTAG. Note that there is no correspondence between the graphs in the two rows.

multiple environments, a condition that is often impractical for real-world graph data. Additionally, some multi-source-free domain adaptation methods can be easily applied to graphs (Dong et al., 2021; Li et al., 2024b; Shen et al., 2023); however, these methods require the use of target data in the model training process. In contrast, the source-free graph model generalization method proposed in this work presents a more viable solution with broader practical implications.

## D.2 MODEL MERGING AND MoE

Model merging and MoE are two techniques for reusing pre-trained models to construct aggregation systems with enhanced performance or generalization capabilities (Yadav et al., 2024). Model merging (Zheng et al., 2023a) typically involves the fusion of model parameters, such as linear averaging (Utans, 1996; Wortsman et al., 2022), task arithmetic merging (Ilharco et al., 2022), or integration based on hidden representations (Yang et al., 2023; Matena & Raffel, 2022; Stoica et al., 2023). These methods are primarily applied to vision and language models, which share consistent architectures that allow parameter space operations. However, such approaches are rarely applied to graph models due to their unique structures. Consequently, the MoE framework (Shazeer et al., 2017) has gained more attention in the graph learning field. In general, MoE facilitates fine-grained fusion of expert outputs, such as (Liu et al., 2023; Wang et al., 2024; Zeng et al., 2023; Liu et al.,

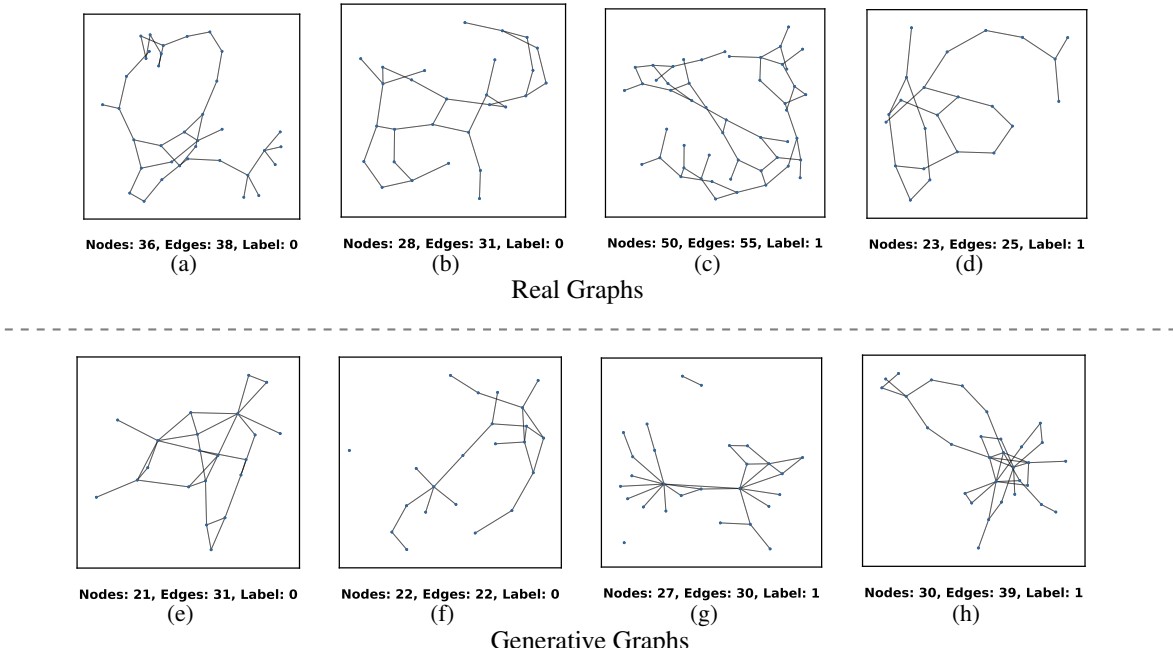

Figure 19: Graph visualization on NCI1. Note that there is no correspondence between the graphs in the two rows.

2024; Ma et al., 2024; Wu et al., 2024b; Liu et al., 2025; Cai et al., 2025). However, these works do not address the source-free out-of-distribution graph models merging we tackle. In this paper, we leverage the MoE framework as a mechanism to select and reuse graph models effectively to achieve domain knowledge fusion.

### D.3 MASK TUNING

Mask tuning is a simple yet effective fine-tuning strategy (Zhao et al., 2020; Radiya-Dixit & Wang, 2020), where a mask matrix is learned for specific modules of a pre-trained model to cover parameters, thereby avoiding redundant computations during the fine-tuning process. This approach originates from model pruning, which uses binary masks to identify important and sparse parameters (Lin et al., 2020; Csordás et al., 2020). In multi-task problems, RMT (Zheng et al., 2023b) applies this strategy to facilitate transfer learning in vision-language models under the zero-shot setting. Similarly, GMT (Li et al., 2024a) leverages gradient information to identify key network parts for sparse updates. Regarding efficient utilization of parameter gradients, (Wang et al., 2022) introduces a judgment criterion to measure the trends of parameters across modules during continual learning, which inspired our research on mask locations. However, there has been limited exploration of mask tuning in graph models. Unlike vision or language models, graph models typically have fewer layers, and the impact of masks on pre-trained GNNs requires further investigation.

### E LIMITATIONS AND FUTURE WORK

**Generalization on More Diverse Graph Data:** Our work is based on the assumption of a mixture distribution, which has been extensively applied in multi-domain generalization problems. For graph data, we have verified this assumption both theoretically and experimentally within the context of graph-level classification tasks. However, the discrepancy among graph domains can be complex, and significant biases can exist across different tasks, graph parametric representations, and scenarios. This variability poses a challenge to the development of graph foundation models (Fu et al., 2024). For the same reason, our approach may not generalize well enough to unknown scenarios, like those with new classes. In future work, we aim to further explore how to extend the multi-task learning capabilities of our model and adapt it to more diverse graph data.

**Towards Scaling Law:** Additionally, the generalization performance of the proposed method is contingent on multiple factors, including the in-distribution performance of pre-trained models, the impact of fine-tuning methods, and the inherent randomness in generative graphs. While the overall computational complexity is relatively low, finding the optimal fitting function remains a challenging task. As the model collection grows, integrating a larger number of more diverse and heterogeneous experts may become a significant hurdle for MoE-based techniques (He, 2024). Consequently, future efforts will focus on investigating model-centric scaling laws for graph expert systems, such as exploring how generalization performance evolves as the number / diversity / capacity of experts grow, and developing automated expert selection and routing mechanisms that can efficiently manage large-scale heterogeneous model pools.

**Future Directions:** In this work, we not only address the novel challenge of model generalization for graphs but also highlight several promising directions for future research: (1) *Extension to Cross-Task Transfer Learning*: Expanding our approach to cross-task transfer learning by integrating and selecting graph models trained on different objectives. This will enable broader applicability of domain generalization across various graph-related tasks. (2) *Model Reuse for Feature / Structural Heterogeneity*: Investigating solutions for model reuse that can effectively handle feature and structural heterogeneity across different graphs. This would enhance the adaptability of pre-trained models to diverse graph characteristics. (3) *Building High-Quality Graph Model Pools*: Researching methods for constructing high-quality graph model pools along with effective ranking and selection strategies. This will facilitate efficient adaptation of graph foundation models to new datasets and domains, similar to the successful adaptation in other areas of machine learning.

