# OpenReview forum: "Out-of-Distribution Graph Models Merging"
_ICLR.cc/2026/Conference — ICLR 2026 Poster_

### Official Review · Reviewer_GZwp · 2025-10-27

**Soundness:** 2
**Presentation:** 2
**Contribution:** 2
**Rating:** 4
**Confidence:** 4

**Summary:**

This paper tackles the problem termed Out-of-Distribution Graph Models Merging, aiming to build a generalized graph model by merging multiple pre-trained GNNs trained on different domains, without requiring access to source data. The authors propose a two-stage framework: (1) a label-conditional graph generation phase that inverts each pre-trained GNN to synthesize pseudo-graphs representing its domain knowledge, and (2) a model merging phase that employs a fine-tuned Mixture-of-Experts with learnable masks and sparse gates to integrate the knowledge of heterogeneous GNNs. Theoretical analyses are provided based on a mixture distribution assumption, and experiments o graph classification benchmarks show improvements over ensemble and model soup baselines.

**Strengths:**

1. The idea of merging multiple pre-trained graph models without source data is interesting and practically relevant in the context of model reusability and privacy-aware learning.

2. The combination of graph generation and MoE-based merging is conceptually coherent and could inspire further exploration of source-free graph model integration.

3. The paper is well-organized and provides technical clarity, including equations, regularizers, and ablation settings.

**Weaknesses:**

1. The proposed task largely overlaps with graph-free knowledge distillation [1] and model soup [2], making the originality less substantial. The contribution seems more like a synthesis of existing ideas than a fundamentally new paradigm.

2. The mixture distribution assumption $G_{T} = \sum_{i} \alpha_{i} G_{i}$ is overly strong and lacks empirical justification, especially in discrete graph spaces. Theoretical analysis is abstract and disconnected from the implemented MoE mechanism.

3. All experiments are conducted on small, low-diversity datasets. No results are shown on large-scale or node-level tasks (e.g., OGB benchmarks). Hence, the claimed “OOD generalization” may not hold in realistic settings.

4. The “domain” split based solely on edge-to-node ratio may not meaningfully reflect true distribution shifts. The evaluation therefore risks testing within-distribution variations rather than genuine OOD generalization.

5. There is no investigation into how experts or masks behave, how gating decisions distribute across domains, or what knowledge is actually merged.

6. Competing baselines are mainly adapted from non-graph domains; missing comparisons to recent source-free domain generalization or prompt-based GNN adaptation methods weakens the empirical argument.

[1]  Graph-free knowledge distillation for graph neural networks.

[2] Model soups: averaging weights of multiple fine-tuned models improves accuracy without increasing inference time.

**Questions:**

1. How sensitive is the method to the number and diversity of pre-trained GNNs? Would OGMM still perform well when experts are highly redundant or of poor quality?

2. How stable is the label-conditional graph generation? Are there cases of mode collapse or low diversity?

3. How does OGMM scale with the number of experts (in terms of computation and performance)?

4. Could the authors provide a qualitative or quantitative analysis of the learned gating and mask patterns to better interpret the merging mechanism?

---

> ### Author Response · Authors · 2025-11-24
> **Author Response to Reviewer GZwp (Weakness 1, 2, 3)**
>
> **Response to W1:**
>
> We want to clarify that our proposed task has fundamental differences from both graph-free knowledge distillation and model soup.
>
> Graph-free knowledge distillation focuses on transferring knowledge from a single teacher model without accessing source data, while model soup merges multiple models through parameter averaging to achieve better performance. In contrast, our task addresses out-of-distribution model merging in a source-free setting, where multiple heterogeneous models trained on different source domains need to be adaptively combined for an unknown target distribution.
>
> The key distinction is that we deal with distribution shift and selective model reuse rather than simple knowledge transfer or parameter averaging. While these existing techniques can be applied to our problem (as we demonstrate in **Table 1**), they fail to capture the core challenge: different source models suffer from varying degrees of negative transfer on the target domain, requiring selective routing rather than uniform combination. Our MoE-based framework specifically addresses this problem, which is fundamentally different from distilling a single model or averaging parameters.
>
> **Response to W2:**
>
> Thanks for your concern. Let us clarify why this assumption makes sense and how it connects to our method.
> The mixture distribution assumption can be commonly found in prior theoretical work [1,2,3,4,5]. Reference [4] proved that one can achieve low loss on target domains by properly combining source distributions, and they argued that different sources should have their own classifiers instead of forcing everything into one unified model. We're following this same logic.
>
> Look at **Figure 2** and **Table 1** - different ways of combining models (Avg-PTM, Ens-Prob, OGMM) all beat single models. This shows the target distribution can reasonably be viewed as a mix of source distributions. Beyond theory, this just reflects how things work in practice. When people face new situations, they don't invent completely new behaviors from scratch - they remix existing patterns. The same happens with graph data. Social network behaviors don't suddenly jump to something totally new; they gradually blend existing patterns. A user might start mixing work-oriented and entertainment-oriented behaviors, creating hybrid patterns that are really just weighted combinations of what already exists.
>
> About the connection with MoE: **Appendix A.2** walks through how we get to Equation 27, which is exactly an MoE formulation.
>
> **Reference**
>
> [1] Blitzer, J., Crammer, K., Kulesza, A., Pereira, F., & Wortman, J. (2007). Learning bounds for domain adaptation. Advances in neural information processing systems, 20.
>
> [2] Crammer, K., Kearns, M., & Wortman, J. (2008). Learning from multiple sources.
>
> [3] Ben-David, S., Blitzer, J., Crammer, K., Kulesza, A., Pereira, F., & Vaughan, J. W. (2010). A theory of learning from different domains. Machine learning, 79, 151-175.
>
> [4] Mansour, Y., Mohri, M., & Rostamizadeh, A. (2008). Domain adaptation with multiple sources. Advances in neural information processing systems, 21.
>
> [5] Xu, R., Chen, Z., Zuo, W., Yan, J., & Lin, L. (2018). Deep cocktail network: Multi-source unsupervised domain adaptation with category shift. In Proceedings of the IEEE conference on computer vision and pattern recognition (pp. 3964-3973).
>
> **Response to W3:**
>
> We have provided experimental results on OGB datasets for both graph classification and node classification tasks in **Appendix C.4**. These results demonstrate that our framework is reliable in realistic settings.

---

> > ### Author Response · Authors · 2025-11-24
> > **Author Response to Reviewer GZwp (continued, Weakness 4, 5, 6)**
> >
> > **Response to W4:**
> >
> > We appreciate this concern. Our domain split is actually well-motivated and follows established works [1][2]. We use graph density because it's a fundamental parametric representation that naturally creates meaningful domain shifts [3].
> >
> > Let me give you a concrete example with molecular datasets like MUTAG. In drug discovery, molecules with different densities represent distinct chemical profiles. Dense molecular graphs often correspond to highly conjugated aromatic compounds that show different pharmacological behaviors compared to sparse aliphatic structures. For MUTAG's mutagenicity prediction, dense aromatic compounds tend to have elevated mutagenic potential due to reactive metabolites or DNA adducts, while sparse linear molecules involve different toxicity pathways like alkylation mechanisms. Pharmaceutical companies constantly face this - models trained on one chemical space fail when applied to chemically distinct molecular families.
> >
> > Another example is spatiotemporal trajectory data. Active users create dense graphs with many connections, while inactive users have sparse patterns. This density difference causes real distribution shifts that break model generalization, which is exactly the problem we address.
> >
> > **Reference**
> >
> > [1] Luo J, Xiao Z, Wang Y, et al. Rank and align: Towards effective source-free graph domain adaptation[J]. arXiv preprint arXiv:2408.12185, 2024.
> >
> > [2] Luo J, Gu Y, Luo X, et al. Gala: Graph diffusion-based alignment with jigsaw for source-free domain adaptation[J]. IEEE Transactions on Pattern Analysis and Machine Intelligence, 2024, 46(12): 9038-9051.
> >
> > [3] Fu D, Fang L, Li Z, et al. What Do LLMs Need to Understand Graphs: A Survey of Parametric Representation of Graphs[J]. arXiv preprint arXiv:2410.12126, 2024.
> >
> > **Response to W5:**
> >
> > Thanks for raising these important questions. We have conducted comprehensive investigations into these aspects.
> >
> > **Figures 4, 7, 8, 9, and 10** analyze how mask positions affect domain adaptation performance. In **Appendix F.1**, we provide low-dimensional visualizations of generated graphs that reveal what domain-specific knowledge experts actually capture. Additionally, **Appendix F.2** includes analysis of gate parameters showing how merging weights distribute across different domains and how the gating mechanism makes routing decisions.
> >
> > These analyses demonstrate that our MoE framework effectively learns to assign target samples to appropriate experts based on their distributional characteristics, and that the mask mechanism successfully preserves domain-specific knowledge during merging.
> >
> > **Response to W6:**
> >
> > We are the first to propose using model merging to solve source-free graph generalization. Existing model merging work mainly focuses on image [1,2] and LLMs [3,4]. Some source-free domain generalization methods [5,6,7] are difficult to adapt to graph models due to the unique nature of graph structures, while prompt-based GNN methods [8,9] require access to source data, which doesn't fit our setting.
> >
> > In our experimental design, we compare against mainstream model soup methods and distillation-based approaches, and validate across different pre-trained GNN architectures and downstream tasks. The experimental results effectively demonstrate the validity of our framework and claims.
> >
> > **Reference**
> >
> > [1] Ilharco G, Ribeiro M T, Wortsman M, et al. Editing models with task arithmetic[J]. arXiv preprint arXiv:2212.04089, 2022.
> >
> > [2] Matena M S, Raffel C A. Merging models with fisher-weighted averaging[J]. Advances in Neural Information Processing Systems, 2022, 35: 17703-17716.
> >
> > [3] Yadav P, Raffel C, Muqeeth M, et al. A survey on model moerging: Recycling and routing among specialized experts for collaborative learning[J]. arXiv preprint arXiv:2408.07057, 2024.
> >
> > [4] Yang E, Shen L, Guo G, et al. Model merging in llms, mllms, and beyond: Methods, theories, applications and opportunities[J]. arXiv preprint arXiv:2408.07666, 2024.
> >
> > [5] Dong J, Fang Z, Liu A, et al. Confident anchor-induced multi-source free domain adaptation[J]. Advances in neural information processing systems, 2021, 34: 2848-2860.
> >
> > [6] Li X, Li J, Li F, et al. Agile multi-source-free domain adaptation[C]//Proceedings of the AAAI Conference on Artificial Intelligence. 2024, 38(12): 13673-13681.
> >
> > [7] Shen M, Bu Y, Wornell G W. On balancing bias and variance in unsupervised multi-source-free domain adaptation[C]//International conference on machine learning. PMLR, 2023: 30976-30991.
> >
> > [8] Fang T, Zhang Y, Yang Y, et al. Universal prompt tuning for graph neural networks[J]. Advances in Neural Information Processing Systems, 2023, 36: 52464-52489.
> >
> > [9] Fu X, He Y, Li J. Edge prompt tuning for graph neural networks[J]. arXiv preprint arXiv:2503.00750, 2025.

---

> > > ### Author Response · Authors · 2025-11-24
> > > **Author Response to Reviewer GZwp (continued, Question 1, 2, 3, 4)**
> > >
> > > **Response to Q1:**
> > >
> > > We investigate this in **Table 4** and **Figure 15**, where we vary GNN architecture types and the number of domains, both of which change the number and diversity of pre-trained GNNs. Results show our method performs stably across these variations.
> > >
> > > Our framework assumes pre-trained GNNs have already achieved reasonable performance on their respective source domains. Under this assumption, the performance upper bound of the merged model is naturally determined by the quality of source domain GNNs. Since we rely entirely on pre-trained models, their quality does influence synthesized data quality and final performance. Our contribution is to explore the capability boundaries of model merging methods under this assumption. Addressing performance degradation when input models are of poor quality or overfitted is an important direction for future work, potentially requiring quality assessment or adaptive weighting mechanisms.
> > >
> > > **Response to Q2:**
> > >
> > > We ensure stable generation through several mechanisms described in **Section 3.2**. We use standard initialization for graphs and constrain the process with two loss functions. For graph structure, we adopt Gumbel-Softmax with a small temperature coefficient to control sampling stability. These designs help the generation process focus on extracting domain knowledge.
> > >
> > > Since graphs are generated conditioned on labels, you can see visualizations of our generated graphs in **Figures 16 and 17**. The low-dimensional visualizations we provide in **Appendix F.1** show that generated graphs successfully capture domain knowledge while maintaining diversity, without mode collapse.
> > >
> > > **Response to Q3:**
> > >
> > > From a computational perspective, OGMM can partially scale to incremental merging scenarios. When new pretrained models arrive, we only need to generate synthetic data for these new models and add it to the existing training set, so the generation stage is modular. For the merging stage, we need to retrain the MoE module, but the cost is manageable because the gate layer and mask parameters are much smaller than generation components. The sparse gating mechanism helps here, since only $k$ experts are activated per sample, computational complexity doesn't grow linearly with the number of models $M$. Storage does scale linearly with $M$, but our mask parameters are lightweight (around 20\% of expert parameters), so adding new models doesn't dramatically increase memory requirements.
> > >
> > > From a performance perspective, we evaluate this in **Figure 15** by partitioning more source domains to increase the number of experts. The results on REDDIT-B and NCI1 show performance changes as expert count varies, with detailed analysis in **Appendix C.5**.
> > >
> > > **Response to Q4:**
> > >
> > > We have added analysis of gate parameter distributions to validate the rationality of expert assignment in **Appendix F.2**. Regarding mask patterns, we already investigate their impact through single-model domain adaptation in **Figures 4, 7, 8, 9, and 10**.
> > >
> > > **We hope that the explanation and further analysis will address your concerns.**

---

> ### Comment · Reviewer_GZwp · 2025-11-28
>
> Thank you to the authors for the detailed responses. However, due to the recent issues on OpenReview, I am currently unable to update my score.

---

### Official Review · Reviewer_YmPD · 2025-11-01

**Soundness:** 3
**Presentation:** 3
**Contribution:** 2
**Rating:** 6
**Confidence:** 3

**Summary:**

This paper addresses a novel problem in the domain of GNNs: Out-of-Distribution Graph Models Merging (OGMM). The authors propose a framework that merges pre-trained GNNs from multiple domains with distribution shifts to create a unified, generalized model. This approach overcomes the challenge of needing to retrain models from scratch by leveraging pre-trained models to preserve specialized domain knowledge. The two-stage process first generates label-conditional graphs using each model and then fine-tunes and merges them using a MoE module. Experimental results show that OGMM outperforms previous methods on multiple datasets, establishing a new state-of-the-art in graph model generalization.

**Strengths:**

1. The paper introduces a novel challenge of merging out-of-distribution graph models without needing to retrain from scratch, which is both practical and impactful for real-world applications where data is scarce.
2. OGMM consistently outperforms previous fusion methods and demonstrates robustness on large-scale datasets like REDDIT-B and NCI1, showing that it can handle diverse graph domains with different GNN architectures.
3. The authors provide a solid theoretical framework for the problem and support it with comprehensive experiments that demonstrate the framework's effectiveness across multiple domains and datasets.

**Weaknesses:**

1. The two-stage process for merging out-of-distribution models involves multiple steps, including fine-tuning and the use of the Mixture-of-Experts (MoE) module. While effective, the overall time complexity of this process could be quite high, especially as the number of pre-trained models and the size of the graphs increase.
2. The experiments primarily focus on datasets such as REDDIT-B and NCI1, which are not necessarily representative of the most commonly encountered graph types in real-world applications. It would strengthen the paper to include additional tests on more widely used or complex datasets, such as social network graphs or biological networks, to better understand the model's applicability across various domains.

**Questions:**

1. See weaknesses.
2. How are these pre-trained GNNs selected? Are they arbitrarily chosen graph models? If not, how is their optimality determined?
3. Negative transfer is a common issue in generalization domains. I would like to know how OGMM avoids this problem, and whether there is any theoretical or empirical evidence supporting this.
4. Some graph-MoE works should be compared and discussed in the experiments.

---

> ### Author Response · Authors · 2025-11-24
> **Author Response to Reviewer YmPD (Weakness 1, 2, Question 2, 3)**
>
> **Response to W1:**
>
> We'd like to clarify that the merging stage actually has quite low complexity. During merging, we only need to optimize the gate layer and mask layers, where the mask layer parameters account for only about 20\% of the total GNN parameters on average. This significantly reduces training difficulty. For detailed complexity analysis, please refer to **Appendix B.4**.
> The runtime comparison for the merging stage is shown in the tables below:
>
> **Table 1. Training Time (seconds) Comparison**
>
> \begin{matrix}
> \hline
> \text{Methods}  & \text{REDDIT-B} & \text{PTC} & \text{MUTAG} & \text{NCI1}\\\\
> \hline
> \text{GCN} & 97.55  & 17.50 & 9.31 & 116.89 \\\\
> \text{GIN} & 112.42 & 19.09 & 9.47 & 175.76\\\\
> \text{GAT} & 144.30 & 14.01 & 25.26 & 206.27 \\\\
> \text{Multi-GFKD} & 51.66 & 49.02 & 30.54 & 51.06\\\\
> \hline
> \text{OGMM} & 41.28 & 8.25 & 8.92 & 45.99 \\\\
> \hline
> \end{matrix}
>
> **Table 2. Testing Time (seconds) Comparison**
>
> \begin{matrix}
> \hline
> \text{Methods}  & \text{REDDIT-B} & \text{PTC} & \text{MUTAG} & \text{NCI1}\\\\
> \hline
> \text{GCN}  & 0.04 & 0.01 & 0.01 & 0.07 \\\\
> \text{GIN} & 0.04 & 0.01 & 0.01 & 0.09 \\\\
> \text{GAT} & 0.06 & 0.01 & 0.01 & 0.08 \\\\
> \text{Uni-Soup} & 0.05 & 0.03 & 0.03 & 0.09 \\\\
> \text{Greedy-Soup} & 0.07 & 0.03 & 0.02 & 0.11 \\\\
> \text{Multi-GFKD} & 0.09 & 0.02 & 0.01 & 0.16 \\\\
> \hline
> \text{OGMM} & 0.10 & 0.07 & 0.05 & 0.25 \\\\
> \hline
> \end{matrix}
>
> **Response to W2:**
>
> Thanks for this suggestion. We have provided evaluations on OGB and Twitch datasets in **Appendix C.4**. Beyond that, we have also added new node classification datasets, with results shown below:
>
> **Table 3. Node Classification Performance on Facebook-100 and Elliptic.**
>
> \begin{array}
> \hline
> \text{Dataset} & \text{Facebook-100} & \text{Facebook-100} & \text{Elliptic} & \text{Elliptic} \\\\
> \hline
> \text{Metric} & \text{Acc} & \text{Pre} & \text{Acc} & \text{Pre} \\\\
> \hline
> \text{Avg-PTM} & 51.45 \pm 0.47 & 46.17 \pm 0.99 & 60.31 \pm 1.45 & 83.05 \pm 1.31 \\\\
> \text{Ens-Prob} &  46.82 \pm 0.07 & 45.01 \pm 0.02 & 77.81 \pm 0.98 & 83.87 \pm 0.63 \\\\
> \text{Ens-HighConf} & 46.66 \pm 0.09 & 44.97 \pm 0.03 & 81.70 \pm 5.54 & 84.98 \pm 1.38 \\\\
> \text{Uni-Soup} & 55.04 \pm 3.75 & 32.08 \pm 13.73 & 82.39 \pm 0.42 & 84.40 \pm 0.24 \\\\
> \text{Greedy-Soup} & 55.51 \pm 1.46 & 49.47 \pm 23.21 & 82.08 \pm 1.30 & 84.38 \pm 0.24 \\\\
> \text{Inverse-X} & 53.47 \pm 4.78 & 50.20 \pm 3.84 &  82.79 \pm 0.16 & 85.22 \pm 0.08 \\\\
> \text{Multi-GFKD} & 54.23 \pm 1.52 & 50.58 \pm 1.27 & 82.65 \pm 0.21 & 85.18 \pm 0.12 \\\\
> \hline
> \text{OGMM} & \mathbf{56.89 \pm 0.05} & \mathbf{52.51 \pm 0.65} & \mathbf{ 82.89 \pm 0.09} & \mathbf{ 85.25 \pm 0.08} \\\\
> \hline
> \end{array}
>
> These experiments demonstrate that our method generalizes well across diverse graph types and tasks, including social networks and biological networks.
>
> **Response to Q2:**
>
> Thanks for this concerning. From a theoretical perspective, stronger pre-trained models naturally lead to better performance. However, in practical scenarios, obtaining optimal models is often infeasible. In our experiments, we use GNNs that have completed standard training and validation on source domains. While these models may have gaps compared to ideal learners, they represent what can be obtained through standard procedures. We provide all source models used in our code, and all baseline methods are evaluated using the same set of models for fair comparison.
>
> **Response to Q3:**
>
> Thanks for this important question. Negative transfer refers to performance degradation caused by irrelevant or conflicting source knowledge. As shown in **Figure 2**, different architectures and source domain models exhibit varying degrees of performance drop on the target domain, being affected by negative transfer to different extents.
>
> From a theoretical perspective, the key insight is that maintaining separate expert classifiers rather than forcing uniform combination naturally reduces negative transfer. Our generalization error bound demonstrates that when the target distribution is a mixture of source distributions, an MoE-based selective routing mechanism achieves lower error than linear combinations, as it avoids forcing all sources to contribute equally regardless of their relevance.
>
> Empirically, our approach addresses negative transfer in two ways. First, we fine-tune source models with generated data from their respective domains, helping them learn more generalizable cross-domain representations. Second, the MoE framework selectively routes and reuses models, allowing each target sample to leverage the most suitable expert while avoiding harmful knowledge from irrelevant sources.
> This explains why OGMM outperforms single-model approaches, parameter fusion methods like Model Soup, and distillation-based methods in **Table 1**.

---

> > ### Author Response · Authors · 2025-11-24
> > **Author Response to Reviewer YmPD (continued, Question 4)**
> >
> > **Response to Q4:**
> >
> > There are indeed many graph MoE works such as [1][2][3][4]. However, these works do not address the source-free out-of-distribution model merging problem we tackle. Our focus is not on designing novel MoE architectures, but rather on leveraging the MoE framework as a mechanism to select and reuse graph models effectively.
> > Using a standard MoE module is sufficient to validate our framework's effectiveness. In our ablation studies, we already compare MoE modules with and without fine-tuning parameters. We will update the paper to better clarify the distinctions and connections with existing graph MoE works, and further emphasize our unique contributions to the model merging problem.
> >
> > **Reference**
> >
> > [1] Wang H, Jiang Z, You Y, et al. Graph mixture of experts: Learning on large-scale graphs with explicit diversity modeling[J]. Advances in Neural Information Processing Systems, 2023, 36: 50825-50837.
> >
> > [2] Ma L, Han H, Li J, et al. Mixture of link predictors on graphs[J]. Advances in Neural Information Processing Systems, 2024, 37: 16043-16070.
> >
> > [3] Wu S, Cao K, Ribeiro B, et al. GraphMETRO: Mitigating Complex Graph Distribution Shifts via Mixture of Aligned Experts[J]. Advances in Neural Information Processing Systems, 2024, 37: 9358-9387.
> >
> > [4] Liu L, Xia X, Xie Q, et al. Enhanced Expert Merging for Mixture-of-Experts in Graph Foundation Models[C]//The Thirty-ninth Annual Conference on Neural Information Processing Systems.
> >
> > **We hope that the explanation and further analysis will address your concerns.**

---

### Official Review · Reviewer_1A5i · 2025-11-01

**Soundness:** 2
**Presentation:** 2
**Contribution:** 2
**Rating:** 4
**Confidence:** 3

**Summary:**

This paper studies a new problem called out-of-distribution graph models merging, which aims to merge multiple pre-trained GNNs from different domains with distribution shifts into a single generalized model. The proposed OGMM is a two-stage framework, leveraging graph generation and a fine-tuned MoE module to enable generalization under graph OOD scenaros. The paper provides theoretical bounds on generalization error, and extensive experiments on multiple datasets demonstrate substantial performance gains over strong baselines.

**Strengths:**

1) The integration of graph generation and MoE-based model fusion is conceptually coherent, enabling domain knowledge transfer at both data and model levels.

2) The mixture distribution assumption and accompanying error bound provide a formal justification for the merging process.

3) The framework is applicable to heterogeneous GNNs, enhancing its generality and practical relevance.

**Weaknesses:**

1) The motivation for merging multiple pre-trained GNNs is not sufficiently justified. It remains unclear why model-level merging is preferable to retraining on aggregated data or simply using the best domain-specific model. No empirical or application-level evidence is given to show that scenarios requiring model-level merging without data access are common or practically constrained.
2) Methodological novelty appears incremental, as the proposed approach largely builds upon existing techniques in graph distillation and mixture-of-experts regularization without a distinct new principle or paradigm.
3) While the paper qualitatively illustrates synthetic graph realism, it does not quantitatively assess their fidelity to the underlying data manifold, leaving the actual extent of domain knowledge recovery uncertain.
4) Notation suffers from inconsistencies such as {G_i}_{i\in M}, and the overall writing is occasionally unclear, making it difficult to follow the theoretical formulation.

**Questions:**

1) How robust is OGMM to the quality of pretrained GNNs? For example, if one model performs poorly or overfits its domain, does the merging process degrade significantly?
2) Why focus masks primarily on classification heads? Have you ablated masking other layers such as message-passing modules and observed differences in generalization?
3) How do you handle varying graph sizes or node/edge feature dimensions across domains during generation and merging? Is there preprocessing involved?
4) Could OGMM be extended to incremental merging, where new pretrained models arrive sequentially?
5) Would adding a contrastive or alignment loss between expert embeddings further stabilize the merging process?

---

> ### Author Response · Authors · 2025-11-24
> **Author Response to Reviewer 1A5i (Weakness 1, 2, 3, 4)**
>
> **We thank the reviewer for the comments. Please find our responses to your concerns below.**
>
> **Response to W1:**
>
> We would like to clarify why model-level merging is necessary in graph learning scenarios.
>
> In many graph applications such as financial transaction networks or healthcare collaboration networks, domain-specific models are trained on sensitive or proprietary data that cannot be shared or aggregated due to privacy regulations, legal constraints, or competitive concerns. The trained models require orders of magnitude less storage than the original graph data, making model sharing feasible while data sharing is not.
>
> Combining multiple large-scale graph datasets significantly increases computational cost. For graphs with millions of nodes and edges across multiple domains, retraining from scratch on aggregated data becomes prohibitive, especially for GNN architectures that scale quadratically or cubically with graph size.
>
> Furthermore, our empirical evidence demonstrates the necessity of model merging. **Figure 2** and **Table 1** show that directly using the best domain-specific pre-trained GNN suffers varying degrees of performance degradation in OOD scenarios. As discussed in **Lines 42-46**, this motivates our model-level merging approach to leverage complementary knowledge from multiple domain-specific models.
>
> We will strengthen the motivation section in the revised manuscript to better describe these practical constraints in graph learning scenes.
>
> **Response to W2:**
>
> Thanks for raising this concern. We would like to clarify the novelty and contributions of our work.
>
> While existing works [1][2][3] study source-free multi-domain generalization on image data, extending this problem to graph data presents unique challenges. Graph domains exhibit structural property variations that create more complex distribution shifts than typical image domain gaps, requiring fundamentally different approaches for model merging.
>
> Beyond the application domain, we make theoretical contributions. We provide a generalization error bound for merged classifiers on mixture distributions, demonstrating that MoE architectures across multiple source domains achieve stronger generalization on target domains. Based on this theoretical foundation, Our OGMM addresses the gap in graph model merging by introducing a framework that combines graph generation with MoE-based merging to specifically tackle domain model merging in graph scenarios. The approach demonstrates consistent superiority over previous distillation and fusion methods across large-scale datasets (REDDIT-B, NCI1, OGB) and different GNN architectures, validating its practical effectiveness for model reusability and source-free generalization.
>
> We will revise the manuscript to better emphasize these theoretical and methodological contributions and distinguish our work from existing approaches.
>
> **References:**
>
> [1] Guo J, Shah D J, Barzilay R. Multi-source domain adaptation with mixture of experts[J]. arXiv preprint arXiv:1809.02256, 2018.
>
> [2] Robey A, Pappas G J, Hassani H. Model-based domain generalization[J]. Advances in Neural Information Processing Systems, 2021.
>
> [3] Tang Y, Wan Y, Qi L, et al. DPStyler: dynamic promptstyler for source-free domain generalization[J]. IEEE Transactions on Multimedia, 2025.
>
> **Response to W3:**
>
> We have conducted manifold visualization experiments comparing original and generated graphs on NCI1 and REDDIT-B datasets. These results are included in **Appendix F.1** of the current manuscript.
>
> **Figure 18** presents the class-wise distribution of real and synthetic data on Domain A in NCI1. The visualization shows that synthetic data closely aligns with the distribution of real data for both classes, indicating that our method successfully extracts domain-specific knowledge embedded in pretrained models. The distributional similarity between synthetic and real data validates that OGMM generates high-quality synthetic graphs that faithfully represent the characteristics of the original data, which is essential for effective knowledge transfer during the merging process.
>
> **Response to W4:**
>
> Thank you for pointing out these issues. We will carefully review and correct notation inconsistencies throughout the manuscript to improve clarity and readability of the theoretical formulation.

---

> > ### Author Response · Authors · 2025-11-24
> > **Author Response to Reviewer 1A5i (continued, Question 1, 2, 3, 4, 5)**
> >
> > **Response to Q1:**
> >
> > We focus on leveraging pretrained GNNs from their respective source domains to construct a more generalizable model. Under this framework, we assume that the pretrained GNNs have already achieved reasonable performance on their source domains. Consequently, the performance upper bound of the merged model is inherently determined by the errors of the source domain GNNs. Since our framework relies entirely on pretrained models, their quality does influence the quality of synthesized data and subsequently the final performance.
> >
> > We have considered this issue in our experimental design. Our experiments employ multiple GNN architectures (GCN, GAT and GIN) and multiple graph domains, using pretrained GNNs that have undergone standard training and validation on their source domains. The pretrained quality of these models naturally varies, and the domain knowledge they capture is inherently inconsistent. Under these realistic conditions, our method still demonstrates stable performance improvements, which indicates a certain degree of robustness to variations in pretrained model quality.
> >
> > Our work focuses on exploring the capability boundaries of model merging methods under the given assumption. Addressing performance degradation when input models are of lower quality or exhibit overfitting is an important and valuable direction for future work, potentially requiring additional quality assessment or adaptive weighting mechanisms.
> >
> > **Response to Q2:**
> >
> > We have thoroughly discussed mask placement in **Section 4.2**, **Appendix C.1**, and **Appendix C.2**. We validated our design through the performance of GNNs fine-tuned with different masking strategies and the gradient changes during continual training. Our analysis demonstrates that fine-tuning only the classification head parameters (approximately 20\% of total parameters) is sufficient to adapt the model to new data while significantly reducing training complexity.
> >
> > **Response to Q3:**
> >
> > Node feature dimensions can be directly obtained from the pretrained model parameters. For graph sizes, in most practical cases, the scale of samples doesn't contain sensitive information and can be reasonably estimated as prior knowledge.
> >
> > In our experiments, the benchmark datasets we used have consistent feature dimensions and sample distributions across domains. This is actually a common setting in graph OOD research and allows us to focus on evaluating the merging approach itself.
> >
> > For cases where domains have different feature dimensions, we can apply standard preprocessing like feature padding or projection layers to align the feature spaces. Our generation process is actually quite flexible here - it samples graph structures from learned distributions, so it naturally handles different graph sizes. We think explicitly extending the framework to handle heterogeneous features across domains would make the method more broadly applicable, and this is something worth exploring further.
> >
> > **Response to Q4:**
> >
> > OGMM can partially scale to incremental merging scenarios. When new pretrained models arrive, we only need to generate synthetic data for these new models and add it to the existing training set. The generation stage is modular in this sense.
> >
> > For the merging stage, we do need to retrain the MoE module, but the cost is manageable because the gate layer and mask parameters are much smaller than the generation components. The sparse gating mechanism helps here - since only $k$ experts are activated per sample, the computational complexity doesn't grow linearly with the number of models $M$.
> >
> > Storage does scale linearly with $M$, but our approach is relatively efficient. The mask parameters are lightweight (around 20\% of classification head parameters), so adding new models doesn't dramatically increase memory requirements compared to other MoE-based methods.
> >
> > **Response to Q5:**
> >
> > Thanks for this important question. We actually tried adding alignment or contrastive losses between expert embeddings, such as using KL divergence to constrain the representations, but found this hurts performance rather than helping.
> >
> > The reason is straightforward: our core idea is to let each target sample be handled by the most suitable source expert based on its distributional characteristics. Forcing experts' outputs closer or farther apart is essentially doing linear combination-style merging, which is exactly the linear combining rule that [1] points out leads to poor performance. The ideal scenario is to maintain the independence and diversity of experts, allowing the routing mechanism to adaptively select, rather than artificially aligning their representation spaces.
> >
> > **References:**
> >
> > [1] Mansour, Y., Mohri, M., & Rostamizadeh, A. (2008). Domain adaptation with multiple sources. Advances in neural information processing systems, 21.
> >
> > **We hope that the explanation and further analysis will address your concerns.**

---

### Official Review · Reviewer_pjDW · 2025-11-01

**Soundness:** 3
**Presentation:** 2
**Contribution:** 3
**Rating:** 6
**Confidence:** 3

**Summary:**

This paper proposes a novel problem of out-of-distribution graph model merging, aiming to merge models pre-trained on graphs from different domains into a model that generalizes under distribution shifts. A two-stage approach is proposed: first, a graph generator is trained to generate synthetic graph data, which is then used in the second stage to fine-tune the MoE module. Experiments demonstrate that the proposed method effectively generalizes to data with distribution shifts.

**Strengths:**

- The proposed problem of Out-of-Distribution Graph Models Merging is novel.
- The proposed method, OGMM, has a solid theoretical foundation. Generating synthetic data to extract domain knowledge makes sense.
- The experiments demonstrate the effectiveness of OGMM and the contributions of each component.

**Weaknesses:**

- The scenarios considered are limited. More results on other node classification graph datasets could be provided. Additionally, the paper only focuses the OOD scenario within a single graph, without considering cross-dataset or cross-domain scenarios.
- The proposed OGMM relies on a mixture distribution assumption, which is not likely to hold in more complex scenarios.
- The proposed OGMM seems to rely on many hyperparameters, some of which significantly impact model performance according to the analysis in the paper. This could limit its applicability in practical settings.
- Although OGMM is theoretically computationally efficient, some experimental results could be provided.

**Questions:**

See weaknesses.

---

> ### Author Response · Authors · 2025-11-24
> **Author Response to Reviewer pjDW (Weakness 1)**
>
> **Response to W1:**
>
> You can find experiments on node classification (ogbg-Arxiv and Twitch) in **Appendix C.4**. Following your suggestion, We have added two more node classification benchmarks, including *Facebook-100* and *Elliptic*. Across these datasets, our method consistently outperforms all baselines under source-data restricted settings. These results further demonstrate the robustness and generality of our framework. We will integrate full tables and analysis in the revised manuscript.
>
> **Table 1. Node Classification Performance on Facebook-100 and Elliptic.**
>
> \begin{array}
> \hline
> \text{Dataset} & \text{Facebook-100}& \text{Facebook-100} & \text{Elliptic}& \text{Elliptic} \\\\
> \hline
> \text{Metric} & \text{Acc} & \text{Pre} & \text{Acc} & \text{Pre} \\\\
> \hline
> \text{Avg-PTM} & 51.45 \pm 0.47 & 46.17 \pm 0.99 & 60.31 \pm 1.45 & 83.05 \pm 1.31 \\\\
> \text{Ens-Prob} &  46.82 \pm 0.07 & 45.01 \pm 0.02 & 77.81 \pm 0.98 & 83.87 \pm 0.63 \\\\
> \text{Ens-HighConf} & 46.66 \pm 0.09 & 44.97 \pm 0.03 & 81.70 \pm 5.54 & 84.98 \pm 1.38 \\\\
> \text{Uni-Soup} & 55.04 \pm 3.75 & 32.08 \pm 13.73 & 82.39 \pm 0.42 & 84.40 \pm 0.24 \\\\
> \text{Greedy-Soup} & 55.51 \pm 1.46 & 49.47 \pm 23.21 & 82.08 \pm 1.30 & 84.38 \pm 0.24 \\\\
> \text{Inverse-X} & 53.47 \pm 4.78 & 50.20 \pm 3.84 &  82.79 \pm 0.16 & 85.22 \pm 0.08 \\\\
> \text{Multi-GFKD} & 54.23 \pm 1.52 & 50.58 \pm 1.27 & 82.65 \pm 0.21 & 85.18 \pm 0.12 \\\\
> \hline
> \text{OGMM} & \mathbf{56.89 \pm 0.05} & \mathbf{52.51 \pm 0.65} & \mathbf{ 82.89 \pm 0.09} & \mathbf{ 85.25 \pm 0.08} \\\\
> \hline
> \end{array}
>
> Additionally, our work is not limited to a single-graph OOD scenario.
> Our main experiments include both **graph classification** and **node classification**, rather than only slicing a single graph.
> For node classification (referring to Appendix C.4), the pre-trained models are obtained from **different temporal snapshots** of citation networks (ArXiv) and **different regional networks** (Twitch), which naturally constitute **cross-dataset / cross-domain** OOD settings. These represent realistic distribution variations across related graphs.
>
> As stated in **Lines 107**, we focus on domains ``that share the same label space $\mathcal{Y}$ as the source data but follow different distributions.'' This corresponds to practical deployment scenarios where the task semantics remain identical while the data distribution shifts, such as temporal drifts in social networks, structural variations in molecular graphs, or the evolving topology of biological systems. Your comment refers to cross-dataset scenarios with substantial semantic differences, which typically require additional mechanisms such as feature alignment or task-level compatibility. While this represents an important future direction, our work intentionally focuses on the fundamental challenge of OOD generalization within the same task domain under strict source-data access limitations, which is already highly impactful in real-world graph applications. For example, in molecular property prediction, models trained on one chemical subspace often fail to generalize to another even when predicting the same property---a classic OOD distribution shift rather than a semantic mismatch.

---

> > ### Author Response · Authors · 2025-11-24
> > **Author Response to Reviewer pjDW (continued, Weakness 2, 3)**
> >
> > **Response to W2:**
> >
> > The mixture distribution assumption can be commonly found in the literature, as seen in [1,2,3,4,5], where [4] introduced the source distribution combining rule and give Theorem indicates "there exists a distribution weighted combining rule that has a loss of at most with respect to any target mixture of the source distributions", which underscores the feasibility of achieving low loss in target domains through a combination of source distributions. Building on this premise, they propose that different source domains should possess their own classifiers rather than employing a single unified classifier across all source and target domains. Our assumption and designed method are aligned with their theoretical framework.
> > Explicitly modeling distributions from graph data is challenging, but empirical evidence can be found in the results in **Figure 2** and **Table 1**--using different combining strategies (e.g., Avg-PTM, Ens-Prob, and OGMM) can all improve the performance, validating the mixture distribution assumption as a reasonable basis.
> >
> > Beyond theoretical foundations, the mixture distribution assumption is grounded in principles of human behavior. From a cognitive psychology perspective, individuals exhibit path dependency and pattern reuse characteristics when encountering new situations. Rather than creating entirely novel response patterns, people tend to invoke and combine existing cognitive schemas and behavioral strategies. This principle naturally extends to graph-structured data representing human interactions and behaviors.
> >
> > In real-world graph data scenarios, this phenomenon manifests clearly in temporal distribution shifts. Consider social networks where user behavior patterns evolve over time: such changes typically do not represent abrupt transitions to entirely new patterns, but rather recombinations of existing behavioral modes. For instance, user interaction patterns may gradually incorporate "entertainment-oriented" elements into previously "work-oriented" behaviors, forming hybrid patterns. These emerging patterns are essentially weighted mixtures of historical behavior patterns rather than completely independent new distributions, directly validating our mixture distribution assumption.
> >
> > **References:**
> >
> > [1] Blitzer, J., Crammer, K., Kulesza, A., Pereira, F., & Wortman, J. (2007). Learning bounds for domain adaptation. Advances in neural information processing systems, 20.
> >
> > [2] Crammer, K., Kearns, M., & Wortman, J. (2008). Learning from multiple sources.
> >
> > [3] Ben-David, S., Blitzer, J., Crammer, K., Kulesza, A., Pereira, F., & Vaughan, J. W. (2010). A theory of learning from different domains. Machine learning, 79, 151-175.
> >
> > [4] Mansour, Y., Mohri, M., & Rostamizadeh, A. (2008). Domain adaptation with multiple sources. Advances in neural information processing systems, 21.
> >
> > [5] Xu, R., Chen, Z., Zuo, W., Yan, J., & Lin, L. (2018). Deep cocktail network: Multi-source unsupervised domain adaptation with category shift. In Proceedings of the IEEE conference on computer vision and pattern recognition (pp. 3964-3973).
> >
> >
> > **Response to W3:**
> >
> > In our experimental setup, the number of generated samples $N$ and the number of source domains $M$ are predetermined prior information, not tunable hyperparameters.
> >
> > In OGMM, we have hyperparameters $\{k, \lambda_{mask}, \lambda_{gate}, \tau, \gamma_v, \gamma_p\}$. $\tau = 0.2$ in Equation 8 controls stable sampling.
> > $\gamma_v = 0.9$ and $\gamma_p = 0.9$ control parameter changes from pretrained models to preserve knowledge. These three hyperparameters are set consistently across different datasets without requiring tuning.
> >
> > Dataset-dependent hyperparameters containing $\{k, \lambda_{mask}, \lambda_{gate}\}$ require dataset-specific adjustment. Our ablation studies (Figures 11-14) demonstrate narrow effective ranges: $k \in [2,3,4]$, $\lambda_{mask} = 0.01$, $\lambda_{gate} \in [1, 10]$.

---

> > > ### Author Response · Authors · 2025-11-24
> > > **Author Response to Reviewer pjDW (continued, Weakness 4)**
> > >
> > > **Response to W4:**
> > >
> > > Thank you for the suggestion. We provide runtime comparisons between OGMM and baselines in generating, training and testing phases in Tables 2-4.
> > >
> > > For generation time, OGMM is significantly faster than Multi-GFKD (2-15× speedup across datasets). Compared to Inverse-X, which is a simplified version of OGMM without graph structure optimization, OGMM requires additional time but delivers better performance.
> > > In the training phase, OGMM achieves the best efficiency, being 2-4× faster than traditional GNN methods and outperforming Multi-GFKD.
> > > Regarding testing time, OGMM is slightly slower than baselines but still operates at millisecond to second scale, which is acceptable for practical applications.
> > >
> > > Overall, OGMM shows good computational efficiency, especially in the training phase.
> > >
> > > **Table 2. Generating Time (seconds) Comparison**
> > >
> > > \begin{matrix}
> > > \hline
> > > \text{Methods}  & \text{REDDIT-B} & \text{PTC} & \text{MUTAG} & \text{NCI1}\\\\
> > > \hline
> > > \text{Inverse-X}  & 2030.95 & 88.07 & 63.17 & 452.02 \\\\
> > > \text{Multi-GFKD} & 4052.60 & 1640.38 & 743.02 & 2043.15 \\\\
> > > \hline
> > > \text{OGMM} & 3448.28 & 134.17 & 126.69 & 827.76 \\\\
> > > \hline
> > > \end{matrix}
> > >
> > > **Table 3. Training Time (seconds) Comparison**
> > >
> > > \begin{matrix}
> > > \hline
> > > \text{Methods}  & \text{REDDIT-B} & \text{PTC} & \text{MUTAG} & \text{NCI1}\\\\
> > > \hline
> > > \text{GCN} & 97.55  & 17.50 & 9.31 & 116.89 \\\\
> > > \text{GIN} & 112.42 & 19.09 & 9.47 & 175.76\\\\
> > > \text{GAT} & 144.30 & 14.01 & 25.26 & 206.27 \\\\
> > > \text{Multi-GFKD} & 51.66 & 49.02 & 30.54 & 51.06\\\\
> > > \hline
> > > \text{OGMM} & 41.28 & 8.25 & 8.92 & 45.99 \\\\
> > > \hline
> > > \end{matrix}
> > >
> > > **Table 4. Testing Time (seconds) Comparison**
> > >
> > > \begin{matrix}
> > > \hline
> > > \text{Methods}  & \text{REDDIT-B} & \text{PTC} & \text{MUTAG} & \text{NCI1}\\\\
> > > \hline
> > > \text{GCN}  & 0.04 & 0.01 & 0.01 & 0.07 \\\\
> > > \text{GIN} & 0.04 & 0.01 & 0.01 & 0.09 \\\\
> > > \text{GAT} & 0.06 & 0.01 & 0.01 & 0.08 \\\\
> > > \text{Uni-Soup} & 0.05 & 0.03 & 0.03 & 0.09 \\\\
> > > \text{Greedy-Soup} & 0.07 & 0.03 & 0.02 & 0.11 \\\\
> > > \text{Multi-GFKD} & 0.09 & 0.02 & 0.01 & 0.16 \\\\
> > > \hline
> > > \text{OGMM} & 0.10 & 0.07 & 0.05 & 0.25 \\\\
> > > \hline
> > > \end{matrix}
> > >
> > > **We hope that the explanation and further analysis will address your concerns.**

---

### Meta-Review · Area_Chair_g32S · 2026-01-02

**Summary:**

The paper suffers from several weaknesses in motivation, methodology, and experiments. The motivation is not sufficiently compelling, and it considers limited scenarios that do not convincingly reflect realistic cross‑domain or cross‑dataset OOD settings. In method, the proposed approach appears somewhat incremental, combining existing ideas from model merging and mixture‑of‑experts without introducing a clearly novel principle, while computational complexity and the practical cost of the two‑stage procedure are insufficiently discussed. Experimentally, the evaluation is limited to small and low‑diversity datasets, relies on sensitive hyperparameters, adopts questionable domain split strategies, and lacks strong baselines, making the empirical support for the claimed benefits unconvincing.

The rebuttal addresses a number of the reviewers’ concerns to varying degrees. Several issues, such as hyperparameter specification, additional datasets, domain split justification, and certain assumptions, are satisfactorily clarified in the new version.
Therefore, I recommend weak accept.

**Reviewer Concerns:**

1. Limited scenarios.
The authors provided additional experiments and explanations to broaden the considered scenarios. However, certain limitations remain, and the overall scope is still somewhat restricted.
Assessment: This concern is partially addressed.

2. Hyperparameter sensitivity.
The authors clarified the specific hyperparameter settings used in the experiments.
Assessment: This response helps resolve the reviewers’ concerns.

3. Computational complexity.
The generation stage in the first phase is time‑consuming, which may raise concerns in practical applications. While the authors discussed efficiency, this issue remains non‑trivial.
Assessment: The concern is partially addressed.

4. Insufficient motivation.
The authors provided a more reasonable and clearer explanation of the motivation in the rebuttal.
Assessment: This response sufficiently resolves the reviewers’ concerns.

5. Limited datasets.
The authors added more experimental datasets in the rebuttal.
Assessment: This adequately addresses the reviewers’ concerns.

6. Assumptions without empirical support.
The authors supplemented the rebuttal with extensive references showing that the assumption has been widely adopted in prior work.
Assessment: This response helps resolve the reviewers’ concerns.

7. Baseline comparison issues.
The authors clarified differences in experimental settings between their method and competing baselines.
Assessment: This explanation is reasonable and acceptable.

8. Domain split strategy.
The authors cited widely accepted prior work using similar domain split strategies and provided intuitive justification.
Assessment: This response is convincing and resolves the concern.

**Reviewer Scores:**

For Reviewer pjDW and YmPD, I think they will keep their scores 6 unchanged.
For Reviewer 1A5i, I think he will keep his score 4 unchanged.
For Reviewer  GZwp, he replied but did not express a clear intention to improve the score. Therefore, I am not sure.
However, the authors' response to his questions is quite comprehensive. I think there is a high probability that it will be rated 6.

---

### Decision · Program_Chairs · 2026-01-26

Accept (Poster)